# Reasoning is Periodicity? Improving Large Language Models Through Effective Periodicity Modeling

**Yihong Dong**[1], **Ge Li**[1,2], **Xue Jiang**[1], **Yongding Tao**[1], **Kechi Zhang**[1], **Hao Zhu**[1], **Huanyu Liu**[1], **Jiazheng Ding**[2], **Jia Li** ♂ [1], **Jinliang Deng**[3], **and Hong Mei**[1,4]

[1]School of Computer Science, Peking University [2]aiXcoder
[3]The Hong Kong University of Science and Technology [4]Advanced Institute of Big Data
dongyh@stu.pku.edu.cn, lige@pku.edu.cn

## Abstract

Periodicity, as one of the most important basic characteristics, lays the foundation for facilitating structured knowledge acquisition and systematic cognitive processes within human learning paradigms. However, the potential flaws of periodicity modeling in Transformer affect the learning efficiency and establishment of underlying principles from data for large language models (LLMs) built upon it. In this paper, we demonstrate that integrating effective periodicity modeling can improve the learning efficiency and performance of LLMs. We introduce FANformer, which adapts Fourier Analysis Network (FAN) into attention mechanism to achieve efficient periodicity modeling, by modifying the feature projection process of attention mechanism. Extensive experimental results on language modeling show that FANformer consistently outperforms Transformer when scaling up model size and training tokens, underscoring its superior learning efficiency. Our pretrained FANformer-1B exhibits marked improvements on downstream tasks compared to open-source LLMs with similar model parameters or training tokens. Moreover, we reveal that FANformer exhibits superior ability to learn and apply rules for reasoning compared to Transformer. The results position FANformer as an effective and promising architecture for advancing LLMs. Our code is available at `https://github.com/YihongDong/FANformer.`.

## 1 Introduction

In recent years, large language models (LLMs) have achieved remarkable progress across various natural language processing tasks, establishing themselves as a cornerstone of modern artificial intelligence [Brown et al., 2020, Zhao et al., 2023, Minaee et al., 2024]. The decoder-only Transformer architecture, in particular, has emerged as the de facto standard for LLM development due to its superior performance and scalability [OpenAI, 2023, DeepSeek-AI et al., 2024a, Groeneveld et al., 2024]. Besides these advancements, Transformer-based models are also known for their immense demand for data and computational resources during training [Kaplan et al., 2020, Hoffmann et al., 2022, Chowdhery et al., 2023]. In comparison, humans are able to accomplish similar learning tasks with far fewer resources. This discrepancy suggests that existing LLM architectures still suffer from low learning efficiency, leaving substantial room for improvement in their ability to extract and generalize the knowledge from data.

Periodicity, characterized by recurring patterns, is a ubiquitous phenomenon in human life and learning processes [Buzsaki, 2006, Lake et al., 2017]. The human brain leverages pattern recognition mechanisms to process information and acquire knowledge efficiently [Zalta et al., 2020, Edalati et al., 2023, Zhan et al., 2018]. However, general network architectures represented by Transformers have potential flaws in periodicity modeling, which could hinder their learning efficiency [Dong et al.,

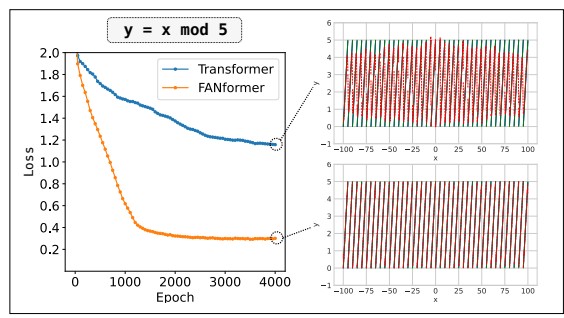
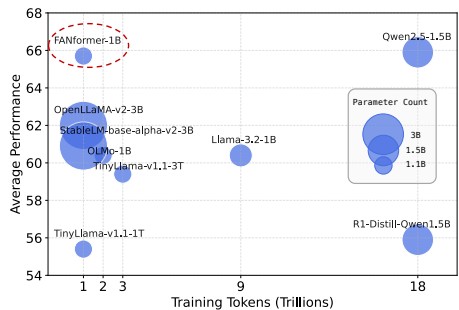

(a) Performance of Transformer and FANformer on periodicity modeling.

(b) Performance of FANformer on language modeling.

Figure 1: The performance of FANformer on periodicity modeling and language modeling. (a) shows the training loss of Transformer and FANformer on the fitting mod function and their performance at the 4,000th epoch. (b) shows the average performance of 8 core commonsense tasks between FANformer-1B and the open-source LLMs with comparable model parameters and training tokens.

2024a, Liu et al., 2020]. As shown in Figure 1 (a), even for a simple mod function, Transformer demonstrates suboptimal performance despite being provided with sufficient training data and model capacity[1]. This inefficiency can be exacerbated during the training process of LLMs to affect their performance, considering the periodicity hidden in large amounts of language data. Fourier Analysis Network (FAN) [Dong et al., 2024a] has shown preliminary success in tasks with explicit or implicit periodic features, but its adaptation to Transformer architectures for large-scale language modeling remains an open challenge.

In this paper, we present FANformer, a novel foundation architecture for LLMs that adapts FAN into the attention mechanism of Transformer to improve learning efficiency and model performance through effective periodicity modeling. It leverages FAN to introduce Fourier principles for capturing and representing periodic patterns, thereby enhancing the Transformer's capability to learn and generalize from data. Specifically, we modify the feature projection process of attention mechanism to incorporate frequency-domain representations to facilitate modeling periodicity. Figure 1 (a) demonstrates the significant advantages of FANformer over Transformer on periodicity modeling, with faster convergence speed and better results. In Figure 1 (b), we can observe that FANformer-1B achieves superior performance with higher utilization efficiency of model parameters and training tokens when benchmarked against comparable Transformer-based LLMs.

To comprehensively validate the effectiveness and scalability of FANformer, we conduct extensive experiments on language modeling tasks. The results of scaling both model parameters and training tokens highlight that FANformer consistently surpasses Transformer, requiring only 69.2% of model parameters or 79.7% of training tokens to achieve comparable performance. We also implement a complete pretraining pipeline to pretrain a 1.1-billion parameter FANformer (FANformer-1B) on 1 trillion tokens. Experiments on various downstream tasks demonstrate that FANformer-1B outperforms open-source LLMs of the same size with fewer training tokens, and exceeds LLMs with three times the parameters when using the same training token. Through further analysis, we reveal that FANformer is a superior choice compared to other variant architectures and discover three interesting findings: 1) By observing the training process, we discover the notable enhancements in FANformer's learning efficiency over Transformer as the model continues to learn from the data. 2) FANformer facilitates the rule-based reasoning paradigm, mitigating the occurrence of "holes" inherent in the case-based learning of Transformer [Hu et al., 2024]. Under the stress test of logical reasoning [Wang et al., 2024], FANformer-1B demonstrates superior performance compared to OLMo-1B and Qwen2.5-1.5B. 3) FANformer's representational capacity consistently surpasses that of Transformer across various layer depths, as evidenced by evaluations of the model's Lipschitz constant [Latorre et al., 2020]. These findings underscore the potential of FANformer as an effective and scalable architecture for advancing LLMs.

The main contributions of our work can be summarized as three points: ❶ We first demonstrate that integrating effective periodicity modeling can improve the learning efficiency and performance of

---

[1]We sample 400K training data from the function of mod 5 and train a 110M Transformer for 4K epochs.

LLMs. ❷ We propose FANformer, a novel LLM architecture, which uses a simple yet effective approach to adapt FAN into attention mechanism for efficient periodicity modeling, consistently outperforming Transformers in scaling model parameters and training tokens. ❸ We pretrain and open-source FANformer-1B, which surpasses SOTA publicly available LLMs with similar parameter counts or training token budgets on downstream tasks.

## 2    Motivation

In this section, we combine formalization with illustrative cases to demonstrate why periodicity modeling facilitates language modeling and reasoning, thereby elucidating the motivation behind the development of FANformer.

The essence of periodicity lies in the repetitive manifestation of certain invariance under transformations, which can be strictly defined through invariance under group actions in Abstract Algebra Dummit and Foote [2004]. Let $X$ be a set and $G$ be a group acting on $X$. An element $x \in X$ is said to be periodic with respect to the action of $G$ if there exists a non-identity element $p \in G$ such that $p \cdot x = x$, where $\cdot$ denotes the group action. The element $p$ is called a period of $x$. Periodicity implies that $x$ is invariant under the action of the cyclic subgroup generated by $p$, denoted by $\langle p \rangle$. For example, $f(a) = f(a + T)$ can be seen as a specific instance of the abstract definition $p \cdot x = x$, where $x = f$, $p = T$, and the group action is translation. When the input $a$ and the group $G$ are extended to higher dimensions or non-temporal domains, the manifestation of the period $T$ also changes accordingly. Crucially, for many reasoning tasks, the underlying operation or inference rule remains invariant across structurally similar subproblems, that is, for all inputs belonging to a certain equivalence class, the same functional rule is applied, which precisely reflects periodic invariance.

Consider addition as an illustrative case: let $f$ represent the addition operation, $a$ denote the digit position index, and let the period $T = 1$ correspond to the positional shift in place value. The reasoning proceeds as:

> *Example: 357 + 286 = ?*
> *Digit-wise operations: units (7 + 6 = 13 → write 3, carry 1); tens (5 + 8 + 1 = 14 → write 4, carry 1); hundreds (3 + 2 + 1 = 6 → write 6).*
> *Result: 643*

Thus, the periodicity of addition is manifested in the repeated application of the same rule across different digit positions, where the rule itself remains invariant under positional shifts. It can be extended to other reasoning, such as logical reasoning, and the scenario is analogous: when a neural network extracts a feature applicable to specific conditions or premises, it repeatedly applies the same invariant rules across analogous subproblems. Such periodic invariance allows for enhancing both the learning efficiency and generalization capability of neural models by reducing redundancy and reinforcing conceptual regularities.

## 3    FANformer

We will provide a detailed description of FANformer for sequence modeling and adopt a decoder-only model to illustrate the architecture.

Given an input sequence $s = \{s_1, s_2, \cdots, s_l\}$, where $s_i$ denotes the $i$-th token and $l$ represents the length of sequence $s$, it is first mapped to the input embedding as $\mathbf{X}^0 = [\mathbf{x}_1, \mathbf{x}_2, \cdots, \mathbf{x}_l] \in \mathbb{R}^{l \times d_h}$, where $d_h$ represents the model's hidden dimension. The embedding is subsequently fed into the model to obtain the final output $\mathbf{X}^N$, with each $n$-th layer of FANformer processing $\mathbf{X}^{n-1}$, where $n \in [1, N]$. The core of each FANformer layer lies in a revised attention module that incorporates a modified FAN layer, referred to as the ATtention-Fourier (ATF) module.

### 3.1    ATF

The attention mechanism serves as a core component of Transformer architectures, enabling dynamic interaction between tokens through query-key-value (QKV) projections. While effective for general sequence modeling, its standard implementation exhibits limitations in capturing periodic patterns due to the inherent locality of linear projections in the time domain. To address this, we propose

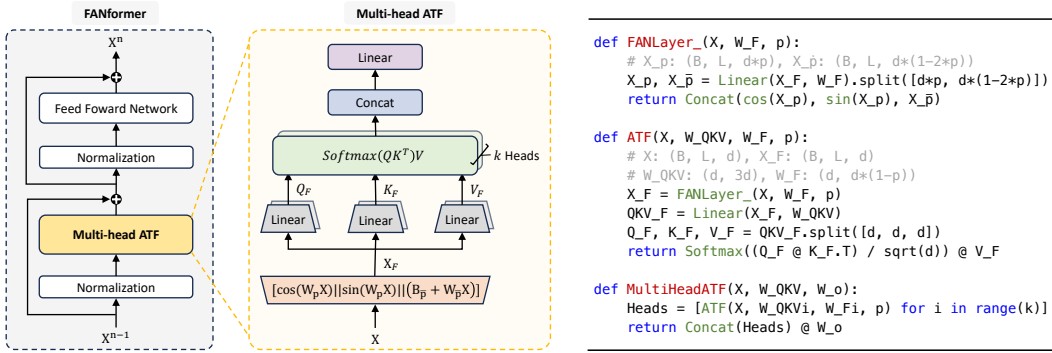

Figure 2: Left: The illustration of FANformer's architecture. Right: The pseudocode of Multi-head ATF, where $p$ is the hyperparameter that controls the proportion of periodicity modeling for $\mathbf{X}_p$.

the ATF module, which incorporates the operations of FAN into the QKV projection process to explicitly model periodicity in the frequency domain. Specifically, given the input $\mathbf{X} \in \mathbb{R}^{l \times d_h}$, we first calculate $\mathbf{X}_F \in \mathbb{R}^{l \times d_h}$ as:

$$\mathbf{X}_F = \text{FANLayer}'(\mathbf{X}) = [\cos(W_p\mathbf{X}) || \sin(W_p\mathbf{X}) || (W_{\bar{p}}\mathbf{X} + B_{\bar{p}})], \tag{1}$$

where FANLayer$'$ represents a variant of the original FAN layer (i.e., Eq. (10)) with the activation function $\sigma$ in Eq. (10) replaced by the identity function, i.e., $\sigma(x) = x$, in this paper, and hyperparameter $p$ is defined as the proportion of $\frac{d_{W_p}}{d_h}$. On this basis, we employ the linear transform to $\mathbf{X}_F$ to compute QKV projections, i.e., $\mathbf{Q}_F, \mathbf{K}_F, \mathbf{V}_F \in \mathbb{R}^{l \times d_h}$, as follows:

$$[\mathbf{Q}_F, \mathbf{K}_F, \mathbf{V}_F] = \mathbf{X}_F[\mathbf{W}_Q, \mathbf{W}_K, \mathbf{W}_V], \tag{2}$$

where $\mathbf{W}_Q, \mathbf{W}_K, \mathbf{W}_V \in \mathbb{R}^{d_h \times d_h}$ are learnable parameters. Similar to the standard attention mechanism, the computation of ATF is defined as:

$$\text{ATF}(\mathbf{X} | \mathbf{W}_Q, \mathbf{W}_K, \mathbf{W}_V) = \text{softmax}\left(\frac{\mathbf{Q}_F\mathbf{K}_F^\top}{\sqrt{d_h}}\right)\mathbf{V}_F, \tag{3}$$

where $\mathbf{Q}_F, \mathbf{K}_F, \mathbf{V}_F$ are computed using the input $\mathbf{X}$ via Eq. (1) and Eq. (2). To enhance the model's capacity, we extend the ATF module to multiple heads. Given input $\mathbf{X} \in \mathbb{R}^{l \times d_h}$, the Multi-head ATF first projects $\mathbf{X}$ into $k$ independent heads through the ATF module. For the $i$-th head, we have:

$$\text{Head}_i = \text{ATF}(\mathbf{X} | \mathbf{W}_Q^i, \mathbf{W}_K^i, \mathbf{W}_V^i), \tag{4}$$

where $\mathbf{W}_Q^i, \mathbf{W}_K^i, \mathbf{W}_V^i \in \mathbb{R}^{d_h \times d_k}$ are learnable parameters for query, key, and value projections respectively, with $d_k = d_h/k$. The outputs of all heads are concatenated and linearly transformed:

$$\text{MultiHeadATF}(\mathbf{X}) = [\text{Head}_1 || ... || \text{Head}_k]\mathbf{W}_O, \tag{5}$$

where $\mathbf{W}_O \in \mathbb{R}^{d_h \times d_h}$ is the learnable parameter of out projection matrix. Note that ATF($\mathbf{X}$) is mathematically equivalent to Attention(FANLayer$'(\mathbf{X})$) (the detailed derivations are provided in Appendix L). This equivalence enables a simple yet effective implementation of Multi-head ATF as shown in Figure 2, which can seamlessly incorporate various advancements in traditional attention mechanisms, such as FlashAttention [Dao et al., 2022].

## 3.2 Overall Architecture

The FANformer model comprises $N$ stacked FANformer layers, where each FANformer layer consists of a Multi-head ATF module and a feed-forward network (FFN) module. Following the previous work [Groeneveld et al., 2024, Touvron et al., 2023a], we employ SwiGLU [Ramachandran et al., 2018, Shazeer, 2020] and pre-norm [Zhang and Sennrich, 2019] as the enhancements to Transformer-based LLMs. Specifically, the $n$-th FANformer layer can be defined as:

$$\mathbf{Y}^n = \text{MultiHeadATF}(\text{Norm}(\mathbf{X}^n)) + \mathbf{X}^n, \tag{6}$$

$$\mathbf{X}^{n+1} = \text{FFN}(\text{Norm}(\mathbf{Y}^n)) + \mathbf{Y}^n, \tag{7}$$

where the MultiHeadATF module is computed via Eq. (5) and the FFN module, which leverages the SwiGLU activation, is expressed as:

$$\text{FFN}(\mathbf{X}) = (\text{Swish}(\mathbf{X}\mathbf{W}_1) \otimes \mathbf{X}\mathbf{W}_2)\mathbf{W}_3, \tag{8}$$

where $\mathbf{W}_1, \mathbf{W}_2 \in \mathbb{R}^{d_h \times d_f}$, $\mathbf{W}_3 \in \mathbb{R}^{d_f \times d_h}$ are learnable parameters, $\otimes$ denotes element-wise multiplication, and $d_f$ is the intermediate dimension. The overview of FANformer's architecture is illustrated in Figure 2.

## 4 Evaluation

We begin with the implementation details of our experiments (Section 4.1), followed by a comprehensive evaluation of FANformer from three distinct perspectives: First, we investigate the scalability of FANformer by examining its performance trends on language modeling tasks with respect to model size and training tokens (Section 4.2). Second, we evaluate the capabilities of the pre-trained FANformer-1B model across multiple downstream tasks (Section 4.3). Third, we conduct an in-depth empirical analysis of FANformer, including ablation study, learning efficiency, reasoning mechanism, representational capacity, and more (Section 4.4). See Appendix A-J for more experiments.

### 4.1 Implementation Details

The experiments are conducted on 80 A100 GPUs. We build FANformer upon the foundation of OLMo [Groeneveld et al., 2024], as it provides a solid pretraining framework of LLMs, with the hyperparameter $p$ set to 0.25 by default. For pretraining FANformer-1B, we randomly sample 1T training tokens from OLMo's training data, i.e., Dolma [Soldaini et al., 2024]. For other experiments, we train models on a smaller sample of Dolma, i.e., Dolma v1_6-sample [AllenAI, 2023], with roughly 10B tokens. The detailed pretraining and experimental setups are provided in Appendix M.

### 4.2 Scalability of FANformer

We explore the scalability of FANformer compared with Transformer to investigate performance trends in the construction of much larger models.

**Setup.** We follow OLMo's configuration and vary the FFN's intermediate dimension $d_f$ to keep the number of parameters consistent for all models in this experiment. For scaling up model parameters, we adopt Dolma v1_6-sample as training data and train LLMs from 268M to 7B. We compare FANformer with the standard Transformer and a variant of FANformer, termed Transformer+ATM, which uses MLP layer instead of FAN layer in FANformer. For scaling up training tokens, we train 1B LLMs on the first 200 billion of our sampled 1T tokens.

**Results.** As shown in Figure 3, the scaling law [Kaplan et al., 2020] empirically aligns well with the results obtained from our FANformer, underscoring its superior scalability properties. Figure 3 (left) reveals that the implementation of FAN consistently surpasses the performance of the standard Transformer across a range of model sizes. This finding highlights FANformer's enhanced scalability in terms of parameter efficiency, as it achieves comparable performance with only 69.2% of the parameters required by the standard Transformer. Notably, the scaling curve of Transformer+ATM closely overlaps with that of the

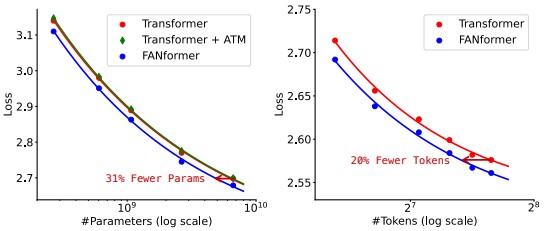

Figure 3: Language modeling loss of scaling up model parameters and training tokens. Left: we train LLMs from 268M to 7B parameters. Right: we evaluate LLMs every 20B tokens up to 200B tokens.

standard Transformer, indicating that merely revising attention mechanisms using MLP Layer is insufficient. This observation further emphasizes that FANformer's performance gains are not attributable to network depth increase, but rather to its special architectural design. Figure 3 (right) demonstrates that FANformer achieves performance parity with the standard Transformer while utilizing significantly fewer training tokens. Specifically, FANformer requires only 159.6B training

Table 1: Zero-shot performance of FANformer-1B versus other comparable open-source LLMs on 8 core tasks from the downstream evaluation suite following OLMo. The results of baselines are taken from the previous works [Groeneveld et al., 2024, Ye et al., 2024, Dong et al., 2024b].

| Models | Param. | Tokens | ARC-C | ARC-E | BoolQ | Hella. | OBQA | PIQA | SCIQ | Wino. | Avg. |
|---|---|---|---|---|---|---|---|---|---|---|---|
| *LLMs around 1B parameters* | | | | | | | | | | | |
| Qwen2.5-1.5B | 1.5B | 18T | 41.2 | 75.5 | 74.0 | 50.2 | 52.4 | 75.7 | 94.7 | 63.3 | 65.9 |
| R1-Distill-Qwen1.5B | 1.5B | 18T+ | 36.2 | 54.4 | 69.1 | 41.8 | 35.4 | 65.1 | 89.5 | 55.3 | 55.9 |
| Llama-3.2-1B | 1.1B | 9T | 31.4 | 65.6 | 64.3 | 47.8 | 46 | 74.5 | 92.3 | 60.7 | 60.4 |
| TinyLlama-v1.1 (3T) | 1.1B | 3T | 34.8 | 53.2 | 64.6 | 58.7 | 43.6 | 71.1 | 90.5 | 58.9 | 59.4 |
| OLMo-1B | 1.1B | 2T | 34.5 | 58.1 | 60.7 | 62.5 | 46.4 | 73.7 | 88.1 | 58.9 | 60.4 |
| *LLMs trained on 1T tokens* | | | | | | | | | | | |
| OpenLLaMA-v2-3B | 3B | 1T | 33.9 | 67.6 | 65.7 | 70.0 | 26 | 76.7 | 92.9 | 62.9 | 62.0 |
| StableLM-v2-3B | 3B | 1T | 32.4 | 67.3 | 64.6 | 68.6 | 26.4 | 76 | 89.5 | 62.1 | 60.9 |
| TinyLlama-v1.1 (1T) | 1.1B | 1T | 33.1 | 49.5 | 58.4 | 52.5 | 37.8 | 70.4 | 86.4 | 55.2 | 55.4 |
| FANformer-1B | 1.1B | 1T | 43.8 | 72.5 | 64.9 | 64.7 | 48.2 | 75.5 | 94.8 | 61.3 | 65.6 |

tokens to match the performance of the standard Transformer trained on 200B tokens, representing a 20.3% reduction in training resource requirements. These findings suggest that FANformer exhibits superior utilization efficiency in terms of both model parameters and training tokens compared to the standard Transformer architecture.

## 4.3 Performance of FANformer-1B

We pretrain FANformer-1B on 1 trillion tokens and report zero-shot performance on a set of commonsense downstream tasks, following previous work [Brown et al., 2020, Touvron et al., 2023b, Groeneveld et al., 2024, inter alia].

**Setup.** The downstream evaluation suite consists of 8 core commonsense tasks, including ARC-C [Clark et al., 2018], ARC-E [Clark et al., 2018], BoolQ [Clark et al., 2019], HellaSwag [Zellers et al., 2019], OBQA [Mihaylov et al., 2018], PIQA [Bisk et al., 2020], SCIQ [Welbl et al., 2017], and WinoGrande [Sakaguchi et al., 2020]. We compare pretrained FANformer-1B to seven open-source LLMs with comparable model parameters or training tokens, including Qwen2.5-1.5B [Team, 2024], R1-Distill-Qwen1.5B [DeepSeek-AI et al., 2025], Llama-3.2-1B [Dubey et al., 2024], TinyLlama-v1.1-1B [Zhang et al., 2024], OLMo-1B [Groeneveld et al., 2024], OpenLLaMA-v2-3B [Geng and Liu, 2023], and StableLM-v2-3B [Tow, 2023].

**Results.** Table 1 presents the evaluation results of our pre-trained FANformer-1B on downstream tasks. It is evident that FANformer-1B surpasses LLMs with comparable parameter sizes, such as Llama-3.2-1B, TinyLlama-v1.1-3T, and OLMo-1B, while utilizing significantly fewer training data. Compared to the base model OLMo-1B, FANformer-1B achieves a relative improvement of 8.8% in the average performance of downstream tasks using only half the training data. On these tasks, FANformer-1B also demonstrates performance comparable to Qwen2.5-1.5B, which is the current SOTA LLM around 1B. For LLMs training on 1T tokens, FANformer-1B even exceeds LLMs with three times the parameters, showing an average relative performance improvement of 6.0-7.9% across all tasks. Moreover, while R1-Distill-Qwen1.5B shows notable improvements in reasoning capabilities based on its reported performance, it exhibits significantly weaker general performance on these commonsense downstream tasks. This observation shows the shortcomings of distillation, highlighting the necessity of the pre-training stage and the importance of research into more efficient model architectures.

## 4.4 Further Analysis

### 4.4.1 Ablation Study and Variant Analysis

**Setup.** We compare FANformer to other variant architectures, including 1) the above-mentioned Transformer+ATM, 2) Transformer+ATL, which use two linear transforms to compute QKV projection, 3) FANformer (original FAN) that employs Eq. (10) (original FAN layer) instead of Eq. (1) in FANformer, 4) Transformer (FFN ← FAN) where the FFN is replaced with FAN [Dong et al., 2024a], and 5) standard Transformer as their ablations.

Table 2: Results of ablation study and variant analysis on LLMs with 1B parameters trained on Dolma v1_6-sample dataset (about 10B tokens). The complete experimental results can be found in Table 8 and Table 9 of Appendix.

| Variants | Param. | Training Loss ↓ | V2 Eval Loss ↓ | V2 Eval PPL ↓ | V3 Eval Loss ↓ | V3 Eval PPL ↓ | DownStream Avg Acc. ↑ |
|---|---|---|---|---|---|---|---|
| Transformer | 1.0× | 2.889 | 3.33 | 30.20 | 3.07 | 24.28 | 53.10 |
| Transformer (FFN ← FAN) | 1.0× | 2.880 | 3.31 | 29.79 | 3.05 | 23.96 | 53.95 |
| Same Parameter | | | | | | | |
| Transformer + ATM | 1.0× | 2.890 | 3.33 | 30.31 | 3.07 | 24.36 | 53.69 |
| Transformer + ATL | 1.0× | 2.882 | 3.31 | 29.68 | 3.05 | 23.97 | 53.46 |
| FANformer (original FAN) | 1.0× | 2.893 | 3.34 | 30.64 | 3.07 | 24.50 | 53.61 |
| FANformer | 1.0× | **2.863** | **3.30** | **29.40** | **3.04** | **23.62** | **55.19** |
| Same Dimension | | | | | | | |
| Transformer + ATM | 1.06× | 2.886 | 3.33 | 30.18 | 3.06 | 24.28 | 52.86 |
| Transformer + ATL | 1.06× | 2.879 | 3.31 | 29.76 | 3.05 | 23.94 | 54.23 |
| FANformer (original FAN) | 1.04× | 2.887 | 3.34 | 30.57 | 3.07 | 24.39 | 53.13 |
| FANformer | 1.04× | **2.856** | **3.29** | **29.22** | **3.03** | **23.47** | **54.88** |

**Results.** From Table 2, we have the following findings: 1) FANformer consistently outperforms other variant architectures in both scenarios of the same parameter and same dimension on all evaluation metrics. 2) The performance of Transformer+ATM and Transformer+ATL is notably inferior to that of FANformer, indicating that the core improvement stems from the ATF module we designed. 3) Although Transformer (FFN ← FAN) yields some improvement, this enhancement is inferior to the gains achieved by FANformer, suggesting that integrating periodicity modeling within attention is more advantageous than FFN on language modeling. 4) Incorporating activation functions such as GELU into the attention mechanism tends to degrade model performance. Specifically, FANformer (original FAN) and Transformer+ATM exhibit weaker performance compared to FANformer and Transformer+ATL, likely because these activation functions suppress certain features, thereby hindering subsequent attention operations.

### 4.4.2 Training Dynamics

We perform a comparative analysis of the loss trends during the training process between our FANformer and Transformer, as illustrated in Figure 4. The experimental results indicate that the loss of FANformer decreases more slowly in the early stages compared to Transformer, which we hypothesize may be due to the initial lack of established periodic modeling. As the training progresses and periodic modeling gradually improves, FANformer demonstrates a faster convergence rate, with its loss decreasing more rapidly than that of Transformer. Intuitively, once the core of semantic knowledge is established, the model's inherent periodic modeling lets new concepts attach to existing representations instead of being learned from scratch, accelerating subsequent learning. This result suggests that as the model progressively learns from the data, the learning efficiency of our FANformer notably surpasses the standard Transformer.

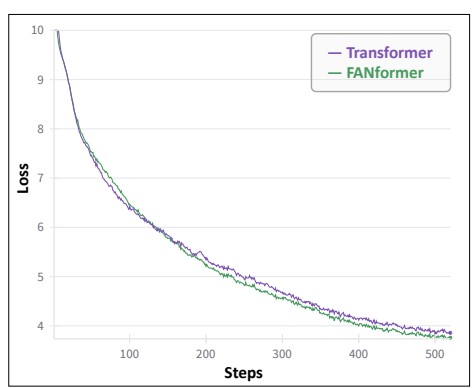

Figure 4: Training loss of FANformer and Transformer on early training steps, with the complete training loss is provided in Figure 7 of Appendix.

### 4.4.3 Case-based and Rule-based Reasoning

**Setup.** Following the work [Hu et al., 2024], we evaluate the case-based and rule-based reasoning of Transformer and our FANformer on two tasks, including: (1) Modular addition: $c = (a + b)$ mod 113 with $a, b \in [0, 112]$; (2) Linear regression: $c = a + 2b + 3$ with $a, b \in [0, 99]$. We finetune pretrained LLMs, i.e., OLMo-1B and FANformer-1B, on each task dataset for 500 epochs and their performance is measured via the Leave-Square-Out method (10 samples per test point).

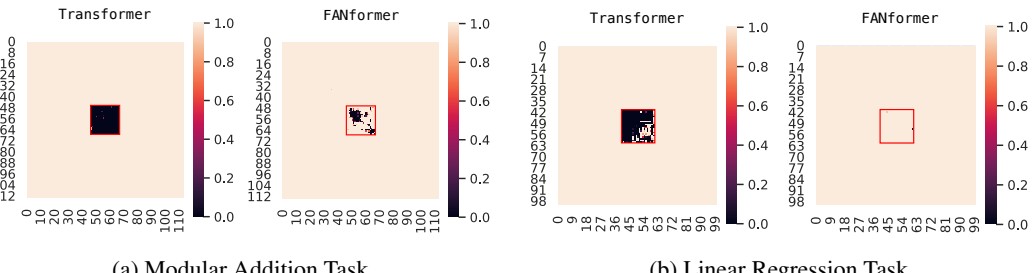

(a) Modular Addition Task          (b) Linear Regression Task

Figure 5: Performance of FANformer and Transformer on modular addition and linear regression tasks, where the darkened regions indicate areas where the model performance approaches zero, signifying the emergence of the "hole" as described in the work [Hu et al., 2024].

**Results.** As shown in Figure 10 of Appendix, both Transformer and our FANformer achieve near-perfect accuracy on the training set of modular addition and linear regression tasks, approaching approximately 100%. However, a critical divergence emerges in their performance on the test sets. Specifically, as shown in Figure 5, Transformer exhibits a pronounced failure to generalize, resulting in a "black hole" in the center of the figure, indicating that its accuracy on the test dataset drops to nearly zero. This observation is consistent with the findings reported in the work [Hu et al., 2024]. In contrast, FANformer demonstrates a marked improvement in addressing the "hole" issue. In the linear regression and modular addition tasks, there is no obvious hole observed, further corroborating the hypothesis that, relative to the Transformer-based model, FANformer possesses a stronger tendency to learn underlying rules, thereby achieving superior generalization performance.

### 4.4.4 LLMs' Proficiency in Logical Reasoning

**Setup.** Following the work [Wang et al., 2024], we adopt the ULogic dataset to systematically evaluate LLMs on their ability to capture underlying inferential logic, where ULogic is constructed after these LLM's training data cutoff, which prevents data contamination [Dong et al., 2024c]. We leverage the Law of Non-Contradiction [Priest et al., 2006], each rule is paired with a negated-conclusion variant; a response is correct only if the model accepts the original and rejects its flip. We evaluate FANformer-1B, OLMo-1B, Qwen2.5-1.5B, and GPT-4 on the two most challenging levels of ULogic for stress-testing.

**Results.** Table 3 summarizes the results on ULogic dataset, with an illustrative case presented in Figure 13 of Appendix. FANformer-1B substantially outperforms both Qwen2.5-1.5B and OLMo-1B: OLMo-1B naively labels every rule as True, and Qwen2.5-1.5B yields largely contradictory answers, whereas FANformer-1B can affirm the original rule and reject its negated counterpart. These results highlight FANformer's superiority in logic reasoning and underscore the architectural advantages conferred by the periodicity-aware design of FANformer.

Table 3: Average performance of different LLMs on ULogic.

| Model | Acc (%) |
|---|---|
| GPT-4 | 65.1 |
| OLMo-1B | 0.0 |
| Qwen2.5-1.5B | 7.1 |
| FANformer-1B | 38.2 |

### 4.4.5 Representational Capacity

**Setup.** We explain the advantage of FAN-former architecture from another perspective: the model's ability to learn complex functions. A larger Lipschitz constant $L$ is frequently linked to greater representational capacity [Latorre et al., 2020, Bartlett et al., 2017], as it enables neural networks to approximate more intricate functional mappings by loosening smoothness constraints in parameter space. This relationship stems from the theoretical framework where less restrictive Lipschitz

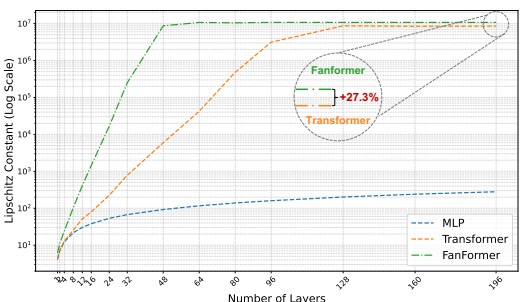

Figure 6: Representational capacity of MLP, Transformer, and FANformer across various layer depths.

conditions permit sharper decision boundaries and richer expressive power. Formally, $L$ satisfies:

$$\forall x, y \in \mathbb{R}^n, \quad \|f_{\text{model}}(x) - f_{\text{model}}(y)\| \leq L\|x - y\|. \tag{9}$$

**Results.**   We evaluate the representational capacity of MLP, Transformer, and FANformer across various layer depths via computing their Lipschitz constant $L$. Although representational capacity rises for all three architectures as additional layers are stacked, the growth rates diverge substantially. FANformer exhibits the steepest trajectory and attains the largest Lipschitz constant $L$ at each depth, consistently surpassing Transformer by 27.3% in deeper configurations. These findings indicate that FANformer has markedly richer expressiveness and a superior ability to model complex functions.

## 5   Related Work

**Large Language Models**   The rapid advancement of LLMs has revolutionized NLP and AI research [Radford, 2018, Dubey et al., 2024, DeepSeek-AI et al., 2025]. The emergence of GPT-3 [Brown et al., 2020], with 175B parameters, showcased remarkable few-shot prompting abilities, suggesting that scaling laws [Kaplan et al., 2020] could unlock emergent capabilities. Recent notable LLMs like PaLM [Chowdhery et al., 2023], LLaMA [Touvron et al., 2023b], GPT-4 [Chowdhery et al., 2023], and DeepSeek [Bi et al., 2024] further pushed the boundaries of model size and performance. Moreover, the open-source release of OLMo [Groeneveld et al., 2024] has provided valuable resources for the community, enabling more accessible training of LLMs.

**Advances in Transformer Architecture**   Recent advancements in Transformer architecture focus on overcoming two main limitations: computational inefficiency in long-context processing and limited expressiveness of attention mechanisms. To address long-context inefficiencies, techniques such as sparsity [Child et al., 2019] and local attention [Beltagy et al., 2020], along with innovations in query mechanisms [Shazeer, 2019, Ainslie et al., 2023] and inference acceleration [DeepSeek-AI et al., 2024b], have been proposed. Hardware-level optimizations like FlashAttention [Dao et al., 2022] further reduce GPU memory access overhead. To enhance the expressiveness of attention mechanisms, methods like Probabilistic Attention Keys [Nguyen et al., 2022] and Selective Attention [Leviathan et al., 2024] improve semantic relationship capture and refine attention by suppressing irrelevant features. Additionally, approaches like Differential Transformer address attention noise in long contexts [Ye et al., 2024]. Different from previous work, we improve language modeling by addressing the challenge of periodicity modeling in Transformers, which can seamlessly incorporate the aforementioned works for revising the attention mechanism, as demonstrated in the derivation provided in Appendix L.

**Fourier-based Networks**   Previous research on Fourier-based Networks was aimed at solving some domain-specific applications [Zuo and Cai, 2005, Tan, 2006, Chen et al., 2022, Li et al., 2021]. Some studies specifically explored the use of sinusoidal activations (e.g., cosine [Silvescu, 1999] [Ngom and Marin, 2021] or sine [Parascandolo et al., 2016, Sitzmann et al., 2020]) to approximate periodic patterns [Liu, 2013]. However, these approaches lacked generalizability beyond narrow domains due to rigid frequency parameterization and limited scalability [Uteuliyeva et al., 2020, Liu et al., 2020]. Recent work [Dong et al., 2024a] addresses these problems using FAN to introduce Fourier Principle into neural networks, but its adaptation to LLMs remains an open challenge. FNet [Lee-Thorp et al., 2022] replaces self-attention with Fourier Transform to achieve linear complexity, but it sacrifices the performance of LMs. In contrast, we employ effective periodicity modeling to improve LLMs.

## 6   Conclusion

We propose FANformer, a novel LLM architecture that enhances learning efficiency by adapting Fourier Analysis Network into attention mechanism for effective periodicity modeling. Experiments show that FANformer outperforms Transformer when scaling model parameters and training tokens, achieving better performance with 31% fewer parameters and 20% fewer tokens. Pretrained FANformer-1B surpasses open-source LLMs of comparable size or training scale on various downstream tasks. The discovery of FANformer's enhanced scalability, learning efficiency, rule-based learning advantages, and representational capacity suggests potential pathways for developing more efficient and high-performance language models.

# 7 Acknowledgement

This research is supported by the National Key R&D Program under Grant No. 2023YFB4503801, the National Natural Science Foundation of China under Grant No. 62192733, 62192730, 62192731, the Major Program (JD) of Hubei Province (No.2023BAA024).

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

# A  Training Loss Curves of OLMO and FANformer

We present the training loss curves for OLMO and FANformer trained on 1 trillion tokens (i.e., 250K steps) in Figure 7.

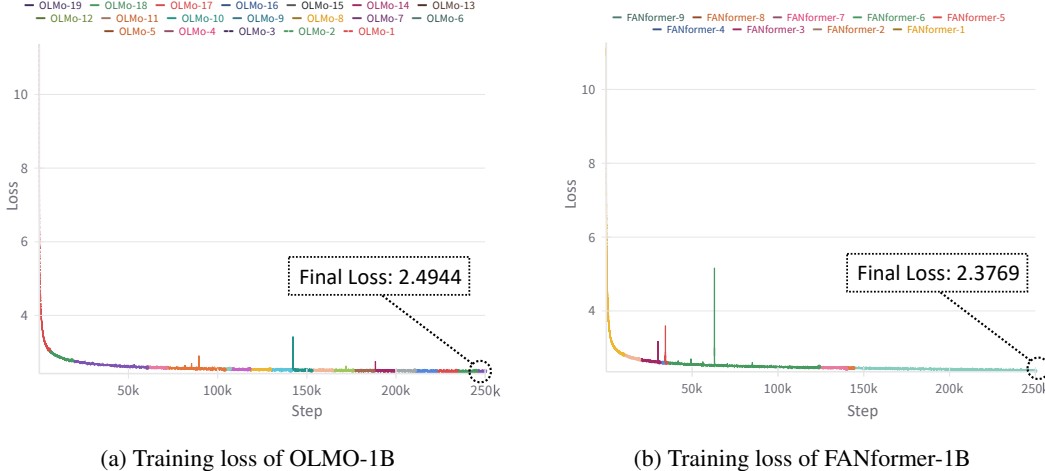

(a) Training loss of OLMO-1B      (b) Training loss of FANformer-1B

Figure 7: The training process of OLMO and FANformer. The data in Figure (a) is sourced from the publicly available results of OLMO (`https://wandb.ai/ai2-llm/OLMo-1B?nw=nwuserdirkgr`).

# B  Preliminary Knowledge

Fourier Analysis Network (FAN) [Dong et al., 2024a] enhances neural networks by introducing Fourier principles for effective periodicity modeling. The core component of FAN is its layer design, which combines periodic basis functions with standard linear transformations. Given a input $\mathbf{X}$, the FAN layer is defined as:

$$\text{FANLayer}(\mathbf{X}) = [\cos(W_p\mathbf{X})\|\sin(W_p\mathbf{X})\|\sigma(W_{\bar{p}}\mathbf{X} + B_{\bar{p}})] \tag{10}$$

where $W_p$ and $W_{\bar{p}}$ are learnable projection matrices, $B_{\bar{p}}$ is a bias term, $\sigma$ denotes an activation function, and $\|$ represents concatenation. Compared to MLP layer, FAN layer explicitly encodes periodic patterns through Fourier series while maintaining general-purpose modeling capabilities.

# C  Experiments on Code Generation and Math tasks

We conduct experiments on code generation tasks (i.e., HumanEval and MBPP) and human-written math tasks (i.e., GSM8K) compared to our baseline, OLMo [Groeneveld et al., 2024]. The results show that our FANformer achieves clear and consistent improvements compared with OLMo on all three benchmarks.

Table 4: Comparison of LLMs on coding and math benchmarks.

| LLMs | Training Tokens | HumanEval | MBPP | GSM8K |
|---|---|---|---|---|
| OLMo-1B | 3T checkpoint | 5.2 | 3.1 | 8.9 |
| FANformer-1B | 1T from scratch | **6.3** | **5.4** | **15.7** |

# D  Effect of hyperparameter $p$

We systematically investigate the impact of hyperparameter $p$, which controls the proportion of periodicity modeling in FANformer, on model performance across its value range. The experimental results from the 1B-scale FANformer (as shown in Figure 8) demonstrate that our model exhibits

strong robustness in terms of training loss and downstream task accuracy, with relatively small performance fluctuations. Furthermore, regardless of the variation in p-values, FANformer consistently outperforms the standard Transformer (horizontal baseline). Analysis of experimental results from models of different scales (300M, 1B, 3B) (as shown in Figure 9) reveals a clear trend: larger models tend to exhibit higher optimal p values. This observation suggests that more powerful FANformers are better equipped to extract more intricate latent periodicity features.

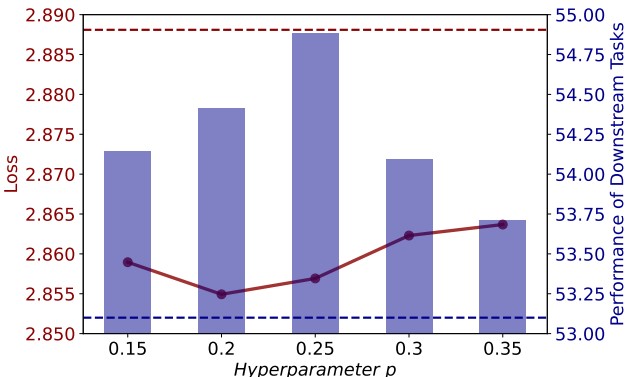

Figure 8: Effect of hyperparameter $p$ in FANformer on its training loss and downstream task performance, where the red dashed line represents the training loss of Transformer, while the blue dashed line denotes the performance on downstream tasks of Transformer.

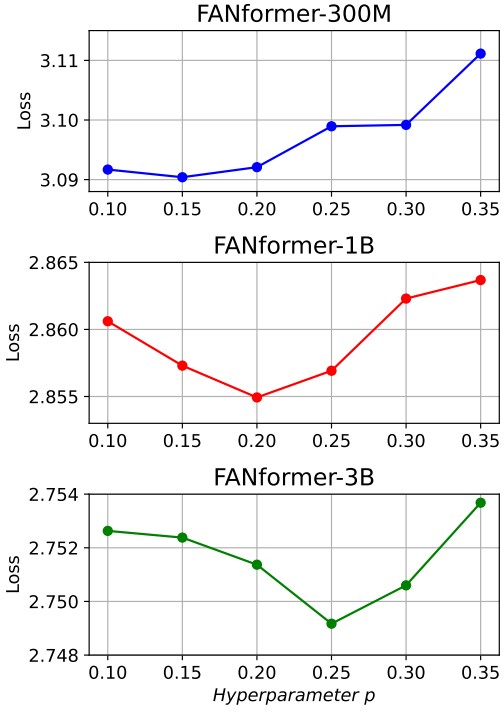

Figure 9: The impact of hyperparameter $p$ on FANformer models of varying sizes.

Table 5: Evaluation results of OLMo-1B-SFT and FANformer-1B-SFT on MMLU, AlpacaEval, ToxiGen, and TruthfulQA (Info+True). Higher values are better for MMLU, AlpacaEval, and TruthfulQA, while lower values are better for ToxiGen.

| Model | MMLU 0-shot ↑ | AlpacaEval %win ↑ | ToxiGen % Toxic ↓ | TruthfulQA (Info+True) Accuracy ↑ |
|---|---|---|---|---|
| **OLMo-1B-SFT** | 24.3 | 1.90 | **2.8** | 55.8 |
| **FANformer-1B-SFT** | **26.7** | **2.51** | 10.4 | **83.0** |

# E  Instruction Following with SFT

## E.1  Models

**FANformer-1B-SFT**: Our pretrained model on 1 trillion tokens, fine-tuned using supervised fine-tuning (SFT) on the `tulu-3-sft-olmo-2-mixture` dataset.

**OLMo-1B-SFT**: A 1B parameter version of OLMo, pre-trained on 3 trillion tokens and fine-tuned using supervised fine-tuning (SFT) on the `tulu-3-sft-olmo-2-mixture` dataset. Model available at `allenai/OLMo-1B-hf`.

For both models, we follow the tokenizer's chat template for prompt formatting when available.

## E.2  Evaluation Setup

We evaluate the models on four benchmarks: MMLU, TruthfulQA, AlpacaEval, and ToxiGen. The evaluation is conducted using the open-instruct evaluation suite, which has been widely adopted for evaluating instruction-tuned language models. Below, we describe the setup for each benchmark.

**MMLU**   We use the official MMLU evaluation script with 0-shot prompting. The maximum token length is set to 2048, and we do not employ chain-of-thought (CoT) reasoning. The evaluation reports the average accuracy across test examples.

**AlpacaEval**   We use the AlpacaEval V1 benchmark with the default setup from the official repository [2]. The evaluated models generate responses for 805 prompts, and GPT-4 is employed to compare the responses with those from the reference model (gpt-4-1106-preview). Given the updates in the AlpacaEval repository, the default models have changed over time. Currently, the default setup uses the weighted_alpaca_eval_gpt4_turbo annotator as the annotator and gpt-4-1106-preview as the reference model. Therefore, our evaluation aligns with the current default configuration in the official AlpacaEval repository.

**ToxiGen**   For ToxiGen, we focus on the prompts specifically designed to elicit toxic language ('hateful' prompts). To reduce evaluation costs, we use a subset of 500 prompts per group for testing. The toxicity classifier used is toxigen_roberta. We report the percentage of generations classified as toxic by the classifier.

**TruthfulQA**   For TruthfulQA, we use the generation setting with the default QA prompt format, including 6 in-context examples. The judge model for evaluating truthfulness and informativeness is `allenai/truthfulqa-truth-judge-llama2-7B`, which is adopted in the `open-instruct` evaluation suite and OLMo's evaluation. We report the percentage of responses deemed both informative and truthful.

# F  Computational Cost Analysis

We analyze the computational overhead of FANformer compared to standard Transformer under two settings: (1) when the number of parameters is matched, the computational cost remains unchanged; (2) when the FFN dimensions remain fixed, the increase in cost is negligible.

---

[2] `https://github.com/tatsu-lab/alpaca_eval`

To ensure a fair comparison, we maintain the same number of parameters in FANformer as in the original Transformer by varying the FFN's intermediate dimension $d_f$ during our experiments. Under this configuration, the computational cost remains equivalent between FANformer (same parameter) and Transformer. We further investigate the setting where FFN dimensions remain unchanged (Table 6). Even for FANformer (same dimension), the computational cost increases only slightly compared to the Transformer. Specifically, the ratio of additional computation can be expressed as:

$$\frac{1.5LSD^2}{L \times (24SD^2 + 4S^2D) + 2SDV} = \frac{1.5D}{24D + 4S} \leq \frac{1.5D}{24D} = 0.0625, \tag{11}$$

where $L$ is the number of layers, $S$ is the sequence length, $D$ is the hidden dimension, and $V$ is the vocabulary size. In practice, the actual additional overhead is much lower than 0.0625, as the term scales inversely with sequence length. This means the overhead diminishes rapidly as sequences grow longer. We summarize the detailed comparison of computational costs between Transformer and FANformer in Table 6.

Table 6: Computational cost of Transformer vs. FANformer ($p = 0.25$).

| Model | Self-Attention/ATF FLOPs | FFN FLOPs | Output FLOPs | Total FLOPs |
|---|---|---|---|---|
| Transformer | $L \times (8SD^2 + 4S^2D)$ | $L \times 16SD^2$ | $2SDV$ | $L \times (24SD^2 + 4S^2D) + 2SDV$ |
| FANformer (Same Param) | $L \times (8SD^2 + 4S^2D + 1.5SD^2)$ | $L \times (16SD^2 - 1.5SD^2)$ | $2SDV$ | $L \times (24SD^2 + 4S^2D) + 2SDV$ |
| FANformer (Same Dim) | $L \times (8SD^2 + 4S^2D + 1.5SD^2)$ | $L \times 16SD^2$ | $2SDV$ | $L \times (25.5SD^2 + 4S^2D) + 2SDV$ |

## G  Inference Speed and GPU Memory Usage

We conduct experiments on the inference speed and GPU memory usage of FANformer relative to Transformer in deployment settings, and have added the experimental results to our revised manuscript. The results show that it add little latency. The configuration of benchmark test: we run for 20 iterations on a single GPU of A100 80G with a fixed sequence length of 4096 tokens and float16 precision.

Table 7: Inference speed and memory usage comparison (Sequence Length=4096).

| Metric | OLMo-1B | FANformer-1B | Difference |
|---|---|---|---|
| Forward Pass Time | 141.49 ms | 142.88 ms | +1.39 ms (+0.98%) |
| Allocated Memory | 4642.69 MB | 4738.86 MB | +96.17 MB (+2.1%) |
| Peak Memory | 6610.70 MB | 6706.88 MB | +96.18 MB (+1.5%) |

## H  Different with Mamba

First, our approach is fundamentally different from SSMs. SSMs model periodicity along the sequence dimension, while our FANformer models it along the feature dimension. Second, our motivation is distinct from that of Mamba. Mamba [Gu and Dao, 2023] is primarily developed to overcome the quadratic computational complexity of Transformers and improve inference efficiency, while our approach is designed to improve learning efficiency and performance of Transformers through effective periodicity modeling.

## I  Detailed Results of Ablation Study for Section 4.4.1

In ablation study, we report the average results across various tasks on V2 Validation Sets, V3 Validation Sets, and Downstream tasks, with the specific tasks detailed in Section M.3. The complete results are detailed in Table 8 and Table 9.

## J  Extended results of Section 4.4.3

The training and testing performance metrics, including loss and accuracy, for case-based and rule-based reasoning are presented in Figure 11 and Figure 12, respectively.

Table 8: The detailed results of ablation study (Part One). All models keep the same number of parameters and are pretrained on Dolma v1_6-sample dataset (about 10B tokens).

| | | | Transformer | Transformer + ATM | Transformer + ATL | FANformer + Activation | FANformer |
|---|---|---|---|---|---|---|---|
| **V2 Validation Sets** | 4chan | Loss | 2.68 | 2.68 | 2.66 | 2.70 | 2.66 |
| | | PPL | 14.60 | 14.53 | 14.36 | 14.88 | 14.34 |
| | c4_100_domains | Loss | 3.11 | 3.11 | 3.10 | 3.12 | 3.08 |
| | | PPL | 22.38 | 22.52 | 22.18 | 22.63 | 21.87 |
| | c4_en | Loss | 3.27 | 3.28 | 3.27 | 3.29 | 3.25 |
| | | PPL | 26.40 | 26.54 | 26.22 | 26.78 | 25.85 |
| | gab | Loss | 3.90 | 3.90 | 3.89 | 3.91 | 3.87 |
| | | PPL | 49.58 | 49.64 | 49.11 | 50.05 | 47.83 |
| | ice | Loss | 3.20 | 3.21 | 3.19 | 3.21 | 3.17 |
| | | PPL | 24.59 | 24.77 | 24.25 | 24.82 | 23.93 |
| | m2d2_s2orc | Loss | 3.56 | 3.57 | 3.56 | 3.59 | 3.56 |
| | | PPL | 35.34 | 35.47 | 34.99 | 36.24 | 35.05 |
| | m2d2_wiki | Loss | 3.14 | 3.14 | 3.13 | 3.15 | 3.11 |
| | | PPL | 23.17 | 23.13 | 22.90 | 23.29 | 22.48 |
| | manosphere | Loss | 3.47 | 3.48 | 3.46 | 3.48 | 3.45 |
| | | PPL | 32.21 | 32.46 | 31.85 | 32.62 | 31.48 |
| | mc4_en | Loss | 3.02 | 3.02 | 3.01 | 3.03 | 2.99 |
| | | PPL | 20.53 | 20.52 | 20.22 | 20.76 | 19.91 |
| | pile | Loss | 2.76 | 2.76 | 2.74 | 2.77 | 2.73 |
| | | PPL | 15.84 | 15.74 | 15.53 | 15.99 | 15.30 |
| | ptb | Loss | 3.68 | 3.70 | 3.64 | 3.71 | 3.66 |
| | | PPL | 39.68 | 40.51 | 38.23 | 40.74 | 38.75 |
| | twitterAEE | Loss | 4.10 | 4.10 | 4.07 | 4.11 | 4.07 |
| | | PPL | 60.25 | 60.18 | 58.79 | 61.10 | 58.54 |
| | wikitext_103 | Loss | 3.33 | 3.33 | 3.30 | 3.35 | 3.29 |
| | | PPL | 28.03 | 28.07 | 27.15 | 28.48 | 26.88 |
| | Average | Loss | 3.33 | 3.33 | 3.31 | 3.34 | 3.30 |
| | | PPL | 30.20 | 30.31 | 29.68 | 30.64 | 29.40 |
| **V3 Validation Sets** | c4_en | Loss | 3.21 | 3.21 | 3.20 | 3.22 | 3.19 |
| | | PPL | 24.80 | 24.86 | 24.60 | 25.04 | 24.24 |
| | dolma_books | Loss | 3.56 | 3.56 | 3.54 | 3.57 | 3.52 |
| | | PPL | 34.98 | 35.32 | 34.43 | 35.57 | 33.96 |
| | dolma_common-crawl | Loss | 3.23 | 3.24 | 3.23 | 3.24 | 3.21 |
| | | PPL | 25.32 | 25.42 | 25.16 | 25.47 | 24.76 |
| | dolma_pes2o | Loss | 2.86 | 2.85 | 2.84 | 2.86 | 2.83 |
| | | PPL | 17.45 | 17.35 | 17.09 | 17.53 | 16.88 |
| | dolma_reddit | Loss | 3.44 | 3.44 | 3.43 | 3.45 | 3.42 |
| | | PPL | 31.13 | 31.35 | 30.94 | 31.42 | 30.54 |
| | dolma_stack | Loss | 1.42 | 1.41 | 1.40 | 1.42 | 1.39 |
| | | PPL | 4.13 | 4.10 | 4.06 | 4.13 | 4.01 |
| | dolma_wiki | Loss | 3.04 | 3.04 | 3.03 | 3.04 | 3.01 |
| | | PPL | 20.89 | 20.84 | 20.62 | 20.97 | 20.26 |
| | ice | Loss | 3.19 | 3.20 | 3.18 | 3.20 | 3.17 |
| | | PPL | 24.41 | 24.56 | 24.09 | 24.63 | 23.75 |
| | m2d2_s2orc | Loss | 3.70 | 3.70 | 3.69 | 3.70 | 3.68 |
| | | PPL | 40.35 | 40.61 | 40.22 | 40.56 | 39.50 |
| | pile | Loss | 2.74 | 2.73 | 2.72 | 2.75 | 2.70 |
| | | PPL | 15.44 | 15.35 | 15.16 | 15.58 | 14.92 |
| | wikitext_103 | Loss | 3.34 | 3.34 | 3.31 | 3.35 | 3.30 |
| | | PPL | 28.21 | 28.21 | 27.33 | 28.57 | 27.03 |
| | Average | Loss | 3.07 | 3.07 | 3.05 | 3.07 | 3.04 |
| | | PPL | 24.28 | 24.36 | 23.97 | 24.50 | 23.62 |
| **Downstream Benchmarks** | piqa | ACC | 66.43 | 66.54 | 65.45 | 66.10 | 66.45 |
| | hellaswag | ACC | 33.87 | 33.84 | 34.02 | 33.75 | 34.37 |
| | winogrande | ACC | 52.80 | 51.62 | 49.96 | 48.78 | 51.72 |
| | openbook_qa | ACC | 28.00 | 28.20 | 28.00 | 28.20 | 29.00 |
| | sciq | ACC | 70.30 | 72.10 | 69.00 | 67.20 | 71.80 |
| | arc_easy | ACC | 45.44 | 46.14 | 47.19 | 47.02 | 45.61 |
| | copa | ACC | 62.00 | 66.00 | 65.00 | 66.00 | 66.00 |
| | rte | ACC | 51.26 | 52.35 | 52.71 | 48.74 | 57.04 |
| | commitment_bank | ACC | 42.86 | 41.07 | 46.43 | 53.57 | 44.64 |
| | mrpc | ACC | 81.05 | 81.22 | 81.22 | 81.22 | 81.47 |
| | sst2 | ACC | 50.11 | 51.49 | 49.08 | 49.08 | 59.11 |
| | Average | ACC | 53.10 | 53.69 | 53.46 | 53.61 | 55.19 |

Table 9: The detailed results of ablation study (Part Two). All models keep the same dimension and are pretrained on Dolma v1_6-sample dataset (about 10B tokens).

| | | | Transformer | Transformer + ATM | Transformer + ATL | FANformer + Activation | FANformer |
|---|---|---|---|---|---|---|---|
| V2 Validation Sets | 4chan | Loss | 2.68 | 2.68 | 2.67 | 2.68 | 2.66 |
| | | PPL | 14.60 | 14.54 | 14.43 | 14.63 | 14.29 |
| | c4_100_domains | Loss | 3.11 | 3.11 | 3.10 | 3.11 | 3.08 |
| | | PPL | 22.38 | 22.43 | 22.11 | 22.49 | 21.69 |
| | c4_en | Loss | 3.27 | 3.28 | 3.26 | 3.28 | 3.24 |
| | | PPL | 26.40 | 26.51 | 26.12 | 26.54 | 25.61 |
| | gab | Loss | 3.90 | 3.90 | 3.89 | 3.91 | 3.87 |
| | | PPL | 49.58 | 49.41 | 48.97 | 50.11 | 47.79 |
| | ice | Loss | 3.20 | 3.20 | 3.19 | 3.21 | 3.17 |
| | | PPL | 24.59 | 24.62 | 24.22 | 24.90 | 23.69 |
| | m2d2_s2orc | Loss | 3.56 | 3.58 | 3.56 | 3.58 | 3.54 |
| | | PPL | 35.34 | 35.73 | 35.17 | 35.78 | 34.58 |
| | m2d2_wiki | Loss | 3.14 | 3.14 | 3.13 | 3.14 | 3.10 |
| | | PPL | 23.17 | 23.04 | 22.81 | 23.10 | 22.27 |
| | manosphere | Loss | 3.47 | 3.48 | 3.46 | 3.48 | 3.45 |
| | | PPL | 32.21 | 32.44 | 31.78 | 32.43 | 31.36 |
| | mc4_en | Loss | 3.02 | 3.02 | 3.01 | 3.03 | 2.99 |
| | | PPL | 20.53 | 20.51 | 20.22 | 20.61 | 19.86 |
| | pile | Loss | 2.76 | 2.76 | 2.74 | 2.77 | 2.72 |
| | | PPL | 15.84 | 15.78 | 15.54 | 15.90 | 15.24 |
| | ptb | Loss | 3.68 | 3.67 | 3.67 | 3.73 | 3.63 |
| | | PPL | 39.68 | 39.19 | 39.15 | 41.67 | 37.82 |
| | twitterAEE | Loss | 4.10 | 4.10 | 4.08 | 4.11 | 4.07 |
| | | PPL | 60.25 | 60.19 | 59.12 | 60.97 | 58.62 |
| | wikitext_103 | Loss | 3.33 | 3.33 | 3.31 | 3.34 | 3.29 |
| | | PPL | 28.03 | 27.96 | 27.29 | 28.22 | 26.98 |
| | Average | Loss | 3.33 | 3.33 | 3.31 | 3.34 | 3.29 |
| | | PPL | 30.20 | 30.18 | 29.76 | 30.57 | 29.22 |
| V3 Validation Sets | c4_en | Loss | 3.21 | 3.21 | 3.20 | 3.21 | 3.18 |
| | | PPL | 24.80 | 24.78 | 24.52 | 24.82 | 24.00 |
| | dolma_books | Loss | 3.56 | 3.56 | 3.54 | 3.56 | 3.52 |
| | | PPL | 34.98 | 35.10 | 34.41 | 35.24 | 33.64 |
| | dolma_common-crawl | Loss | 3.23 | 3.23 | 3.22 | 3.23 | 3.20 |
| | | PPL | 25.32 | 25.25 | 25.09 | 25.35 | 24.55 |
| | dolma_pes2o | Loss | 2.86 | 2.85 | 2.84 | 2.86 | 2.82 |
| | | PPL | 17.45 | 17.37 | 17.12 | 17.44 | 16.79 |
| | dolma_reddit | Loss | 3.44 | 3.44 | 3.43 | 3.44 | 3.41 |
| | | PPL | 31.13 | 31.22 | 30.83 | 31.28 | 30.31 |
| | dolma_stack | Loss | 1.42 | 1.41 | 1.40 | 1.42 | 1.39 |
| | | PPL | 4.13 | 4.09 | 4.07 | 4.13 | 4.02 |
| | dolma_wiki | Loss | 3.04 | 3.03 | 3.03 | 3.04 | 3.00 |
| | | PPL | 20.89 | 20.78 | 20.61 | 20.88 | 20.10 |
| | ice | Loss | 3.19 | 3.20 | 3.18 | 3.21 | 3.16 |
| | | PPL | 24.41 | 24.44 | 24.04 | 24.72 | 23.55 |
| | m2d2_s2orc | Loss | 3.70 | 3.70 | 3.69 | 3.70 | 3.67 |
| | | PPL | 40.35 | 40.50 | 39.99 | 40.56 | 39.17 |
| | pile | Loss | 2.74 | 2.73 | 2.72 | 2.74 | 2.70 |
| | | PPL | 15.44 | 15.39 | 15.17 | 15.50 | 14.87 |
| | wikitext_103 | Loss | 3.34 | 3.34 | 3.31 | 3.35 | 3.30 |
| | | PPL | 28.21 | 28.12 | 27.46 | 28.36 | 27.12 |
| | Average | Loss | 3.07 | 3.06 | 3.05 | 3.07 | 3.03 |
| | | PPL | 24.28 | 24.28 | 23.94 | 24.39 | 23.47 |
| Downstream Benchmarks | piqa | ACC | 66.43 | 65.13 | 66.76 | 66.38 | 66.59 |
| | hellaswag | ACC | 33.87 | 33.96 | 34.22 | 33.92 | 35.15 |
| | winogrande | ACC | 52.80 | 51.62 | 50.12 | 51.07 | 51.38 |
| | openbook_qa | ACC | 28.00 | 28.00 | 28.80 | 28.60 | 28.40 |
| | sciq | ACC | 70.30 | 70.90 | 70.40 | 70.20 | 70.30 |
| | arc_easy | ACC | 45.44 | 48.60 | 47.02 | 44.91 | 48.95 |
| | copa | ACC | 62.00 | 67.00 | 67.00 | 65.00 | 69.00 |
| | rte | ACC | 51.26 | 51.99 | 54.87 | 54.51 | 54.87 |
| | commitment_bank | ACC | 42.86 | 32.14 | 41.07 | 37.50 | 39.29 |
| | mrpc | ACC | 81.05 | 81.17 | 80.59 | 81.17 | 81.11 |
| | sst2 | ACC | 50.11 | 50.92 | 55.73 | 51.15 | 60.55 |
| | Average | ACC | 53.10 | 52.86 | 54.23 | 53.13 | 54.88 |

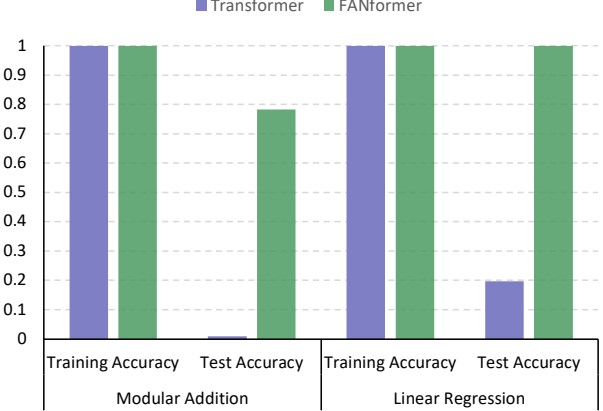

Figure 10: Training accuracy and test accuracy of FANformer and Transformer on modular addition and linear regression tasks.

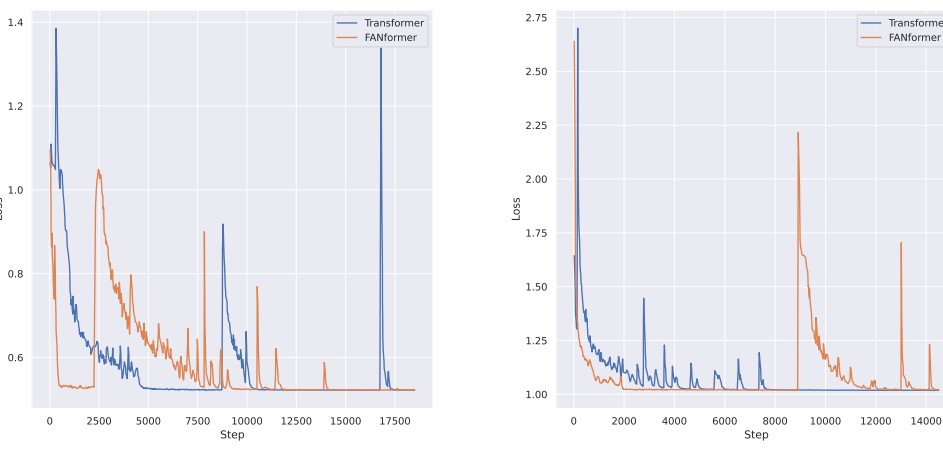

(a) Training loss on modular addition task      (b) Training loss of linear regression task

Figure 11: Training loss of FAN and Transformer on case-based and rule-based reasoning.

## K    Case for Section 4.4.4

We present a case of FANformer and the baselines under a logical reasoning stress-test in Figure 13.

## L    Formal Proof of $\mathbf{ATF}(\mathbf{X}) = \mathbf{Attention}(\mathbf{FANLayer}'(\mathbf{X}))$

**Proposition 1.** *For an input sequence representation* $\mathbf{X} \in \mathbb{R}^{n \times d}$*, the* ATF *operator satisfies*
$$\mathrm{ATF}(\mathbf{X}) = \mathrm{Attention}\big(\mathrm{FANLayer}'(\mathbf{X})\big).$$

*Proof.* By the definition of ATF via Eq. (3), we have
$$\mathrm{ATF}(\mathbf{X}) = \mathrm{softmax}\left(\frac{\mathbf{Q}_F \mathbf{K}_F^\top}{\sqrt{d_k}}\right) \mathbf{V}_F.$$

Substituting $\mathbf{Q}_F, \mathbf{K}_F, \mathbf{V}_F$ from Eq. (2)) into Eq. (3) yields
$$\mathrm{ATF}(\mathbf{X}) = \mathrm{softmax}\left(\frac{(\mathbf{X}_F \mathbf{W}_Q)(\mathbf{X}_F \mathbf{W}_K)^\top}{\sqrt{d_k}}\right)(\mathbf{X}_F \mathbf{W}_V)$$

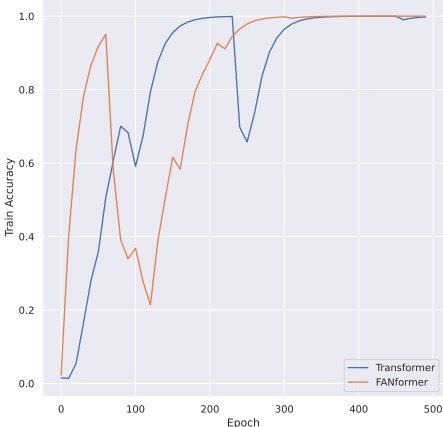
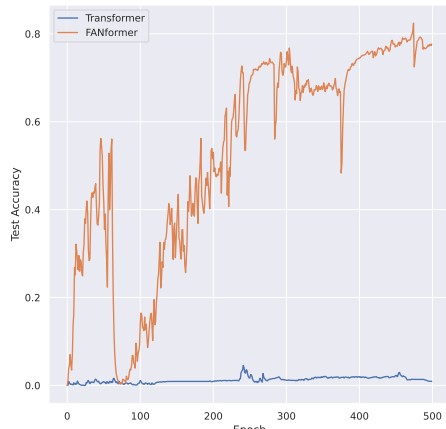

(a) Training accuracy on modular addition task

(b) Test accuracy of modular addition task

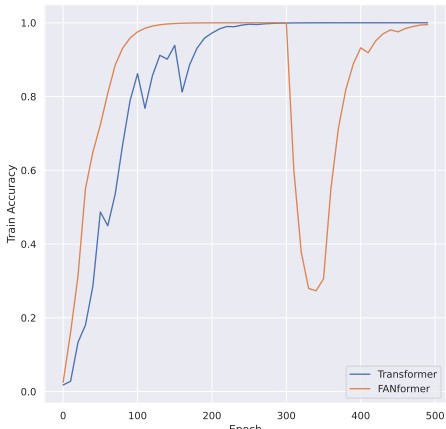
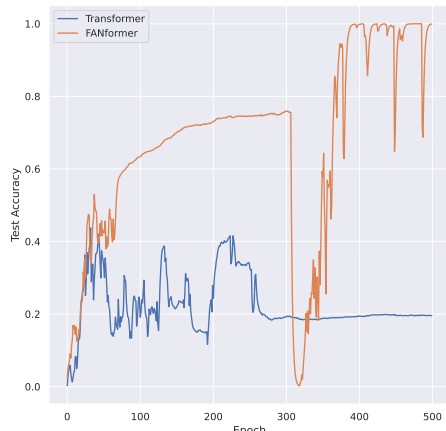

(c) Training accuracy on linear regression task

(d) Test accuracy of linear regression task

Figure 12: Accuracy of FAN and Transformer during training and testing on case-based and rule-based reasoning.

Matching to standard attention, for any input $\mathbf{Z}$, multi-head attention (single head shown for clarity) is defined as

$$\text{Attention}(\mathbf{Z}) = \text{softmax}\left(\frac{\mathbf{Z}\mathbf{W}_Q(\mathbf{Z}\mathbf{W}_K)^\top}{\sqrt{d_k}}\right)(\mathbf{Z}\mathbf{W}_V)$$

Thus, we have

$$\text{ATF}(\mathbf{X}) = \text{Attention}(\mathbf{X}_F). \tag{12}$$

Finally, substituting $\mathbf{X}_F = \text{FANLayer}'(\mathbf{X})$ via Eq. (1) into the above formula establishes the claim:

$$\text{ATF}(\mathbf{X}) = \text{Attention}\big(\text{FANLayer}'(\mathbf{X})\big).$$

$\square$

<table>
<tr><td colspan="2" align="center">Case Study</td></tr>
<tr><td colspan="2">

True or False? Please predict whether the input rule is very likely to be true, and also explain why. Please note that the rule need not necessarily but just very likely to be true.

**Examples**:
*Input: If Show Y was produced at Time Period Z1, Person X died at a Time Period Z2, and Time Period Z1 is earlier than Time Period Z2, then Person X had no chance to access Show Y.*
*Output: False. Because Show Y was available before Person X died.*

*Input: If Person X lives in Region Z and Animal Y inhabits the same Region Z, then Person X can access Animal Y.*
*Output: True. Because person and animal exist in the same region.*

</td></tr>
<tr><td align="center">Original Conclusion</td><td align="center">Flipped Conclusion</td></tr>
<tr>
<td>

**Input**: If Organization X conducts an activity during Time Period Z1 and Technology Y was invented during Time Period Z2, and Time Period Z2 is earlier than Time Period Z1, **then Organization X can access Technology Y**.

**OLMo-1B**

**Output**: True. Because Organization X can access Technology Y.

**Qwen2.5-1.5B**

**Output**: True. Because person and animal exist in the same region.

**FANformer**

**Output**: True. Because technology was invented before the activity was conducted.

</td>
<td>

**Input**: If Organization X conducts an activity during Time Period Z1 and Technology Y was invented during Time Period Z2, and Time Period Z2 is earlier than Time Period Z1, **then Organization X cannot access Technology Y**.

**OLMo-1B**

**Output**: True. Because Organization X cannot access Technology Y.

**Qwen2.5-1.5B**

**Output**: True. Because person and animal exist in the same region.

**FANformer**

**Output**: False. Because Technology Y was invented before Organization X conducts the activity.

</td>
</tr>
</table>

Figure 13: Case of FANformer and the baselines under logical reasoning stress-test.

# M    Comprehensive Experimental Details

## M.1    Detailed training settings of FANformer

We train FANformer-1B using the ZeRO optimizer strategy [Rajbhandari et al., 2020] via PyTorch's DDP framework [Li, 2018]. Following OLMo [Groeneveld et al., 2024], we use a constant global batch size of approximately 4M tokens (2048 instances, each with a sequence length of 2048 tokens). To improve throughput, we employ PyTorch's amp module with the bfloat16 format. We employ the AdamW optimizer [Loshchilov and Hutter, 2019] for the model's training process. The learning rate for all LLMs is set to 4.0e-4. We warm up the learning rate over 2000 steps ( 8B tokens) and then decay it in a cosine manner from there down to a tenth of the peak learning rate over the remainder of training. We employ FlashAttention [Dao et al., 2022] to accelerate the model training and inference processes, leveraging its ability to optimize memory usage and computational efficiency. The total GPU computational cost for pre-training FANformer-1B amounts to approximately 47,600 GPU hours.

## M.2    Detailed Setup for Section 4.2

For different model sizes in Figure 3, the hidden dimension, number of layers, and number of heads are listed in Table 10.

## M.3    Validation Set And Downstream Tasks

Following [Zhu et al., 2024], we use V2 Validation Sets, V3 Validation Sets, and Downstream tasks to evaluate our approach. The specific tasks included in V2 validation sets, V3 validation sets, and downstream tasks are listed in Table 11.

Table 10: Model size and setup used for FANformer in Section 4.2, where Transformers follows the setups of previous work OLMo [Groeneveld et al., 2024].

| Model | Size | Hidden Dim. | Num Layers | Num Heads | Weight Tying |
|---|---|---|---|---|---|
| FANformer-300M | 268M | 1024 | 16 | 16 | True |
| FANformer-600M | 604M | 1536 | 16 | 16 | True |
| FANformer-1B | 1.1B | 2048 | 16 | 16 | True |
| FANformer-3B | 2.6B | 2560 | 24 | 20 | False |
| FANformer-7B | 6.7B | 4096 | 24 | 32 | False |

### M.4    Detailed Setup of Case-based and Rule-based Reasoning.

Following the work [Hu et al., 2024], we focus on binary operations that take two numbers, $a$ and $b$, as inputs. Denoting $c$ as the target label, the constructed datasets are in the form of $\mathcal{D} = \{((a_i, b_i), c_i)\}$ for two mathematical tasks: modular addition and linear regression. The two tasks are defined as follows:

- **Modular addition.** The input to the model is "$a + b =$", and the output is $c$, where $c = (a + b) \mod P$. The values of $a$ and $b$ range from 0 to 112. The constant $P$ is 113 here.

- **Linear regression.** This task involves the model learning a linear regression function. The input is given by "$(a, b) =$", and the output is $c$, where $c = m \cdot a + n \cdot b + p$. The values of $a$ and $b$ range from 0 to 99. The constants are set as $m = 1$, $n = 2$, and $p = 3$.

**Leave-Square-Out**    The work [Hu et al., 2024] employs the Leave-Square-Out method to evaluate the generalization ability of the Transformer [Dong et al., 2024c]. In this approach, a square test set is created to isolate the test samples from the training samples. For instance, consider the center of the square at $(a_k, b_k)$ with a side length of $l_k$. The square test set is defined as $\mathcal{T}_k = \{((a_i, b_i), c_i) \mid a_k - \frac{l_k}{2} \le a_i \le a_k + \frac{l_k}{2}, b_k - \frac{l_k}{2} \le b_i \le b_k + \frac{l_k}{2}\}$, and all remaining samples from the training set. This division creates a "hole" in the center of the training set, which is more challenging for the model compared to a random split. Since there are no similar cases in the training set to aid the model in solving the problem, this method tests whether the model has truly learned the underlying rules. In the experiments of the work [Hu et al., 2024], they found that Transformer-based models fail to generate correct answers for the test set in the "hole". Therefore, we use this method to assess the generalization ability of FANformer.

**Settings**    We finetune both the Transformer and FANformer models on each dataset for 500 epochs. The batch size is set to 336, and the learning rate is initialized at $10^{-4}$. A warm-up ratio of 0.01 is used, and we apply cosine decay to adjust the learning rate throughout the training process.

During generation, we set the model temperature to 0.5 and sample 10 generations to evaluate the accuracy on each test point. The square center $(a_k, b_k)$ is (50, 50) for linear regression and (56, 56) for modular addition.

Following the work [Hu et al., 2024], we apply the Leave-Square-Out method to each dataset. Specifically, we extract a square comprising 441 samples (from a total of approximately 10,000 samples) with a side length of 20 to form our test set, leaving the remainder as the training set. It is important to note that, despite removing a small portion of training samples, we ensure that all tokens present in the dataset appear in the training set. This precaution is to prevent the models from failing simply due to encountering unseen tokens. We then proceed to finetune Transformer and FANformer models using this specific training-test split for each dataset.

### M.5    Assessing LLMs' Proficiency in Capturing Inferential Rules

**Analysis Setup**    Following the experimental setup proposed by Wang et al. [2024], we adopt the ULogic framework to systematically evaluate LLMs on their ability to capture underlying inferential logic. Specifically, we leverage a curated probing subset comprising 1,104 diverse rules drawn from their rule base. These rules—manually verified by the original authors—span a range of lengths,

Table 11: Validation Set And Downstream Tasks.

| **V2 Validation Sets** |
|---|
| v2-small-4chan-validation |
| v2-small-c4_100_domains-validation |
| v2-small-c4_en-validation |
| v2-small-gab-validation |
| v2-small-ice-validation |
| v2-small-m2d2_s2orc-validation |
| v2-small-m2d2_wiki-validation |
| v2-small-manosphere-validation |
| v2-small-mc4_en-validation |
| v2-small-pile-validation |
| v2-small-ptb-validation |
| v2-small-twitterAEE-validation |
| v2-small-wikitext_103-validation |
| **V3 Validation Sets** |
| v3-small-c4_en-validation |
| v3-small-dolma_books-validation |
| v3-small-dolma_common_crawl-validation |
| v3-small-dolma_pes2o-validation |
| v3-small-dolma_reddit-validation |
| v3-small-dolma_stack-validation |
| v3-small-dolma_wiki-validation |
| v3-small-ice-validation |
| v3-small-m2d2_s2orc-validation |
| v3-small-pile-validation |
| v3-small-wikitext_103-validation |
| **Downstream Benchmarks** |
| piqa [Bisk et al., 2020] |
| hellaswag [Zellers et al., 2019] |
| winogrande [Sakaguchi et al., 2020] |
| openbook_qa [Mihaylov et al., 2018] |
| sciq [Welbl et al., 2017] |
| arc_easy [Clark et al., 2018] |
| copa [Roemmele et al., 2011] |
| commitment_bank [De Marneffe et al., 2019] |
| mrpc [Dolan and Brockett, 2005] |
| rte [Dagan et al., 2005] |
| sst2 [Socher et al., 2013] |

polarities, and structural patterns, ensuring broad coverage and high quality. The evaluation is framed as a binary entailment classification task, where the model must determine whether a given rule expresses a valid logical entailment. We employ a two-shot Chain-of-Thought (CoT) prompting method [Wei et al., 2022], in which each input includes one correct and one incorrect example to minimize label bias. The model is prompted not only to make a binary judgment but also to justify its reasoning, with an appended instruction such as "and also explain why."

To further enhance the reliability of the evaluation, we incorporate the Law of Non-Contradiction [Priest et al., 2006], which posits that statements of the form "If X, then Y" and "If X, then not Y" cannot simultaneously be true. Accordingly, for each original rule, we construct a flipped version by negating its conclusion. A rule is considered correctly classified only if the model affirms the original rule as true and rejects the flipped version as false, as illustrated below. We evaluate the base model of FANformer-1B, OLMo-1B, and Qwen2.5-1.5B, as well as GPT-4 on the two most challenging levels of ULogic for stress-testing (i.e., Length 3 and Length 4).

| | |
|---|---|
| *If Premise, then Conclusion_original.* | True |
| *If Premise, then Conclusion_flipped.* | False |

# N  Limitations

Our work has several limitations, which we aim to address in our future work:

First, due to constraints in computational resources, we only pretrain the FANformer-1B on 1 trillion tokens. However, our experimental results regarding FANformer's scalability indicate that our FANformer demonstrates favorable scaling behavior during training, suggesting that increasing the model size and training tokens could lead to more significant performance improvements. To explore this further, we plan to seek additional computational resources to train larger-scale language models.

Second, our work is orthogonal to the existing approaches for revising the attention mechanism, i.e., our work can seamlessly incorporate them, as verified in the derivation provided in Appendix L. There are numerous variants of attention mechanisms, as discussed in the related work (Section 5), such as Flash Attention [Dao et al., 2022], MQA [Shazeer, 2019], and MLA [DeepSeek-AI et al., 2024b]. In this work, we only incorporate Flash Attention for necessary acceleration, while leaving the exploration of other approaches for future work.

Third, although we have observed that enhancing the ability of language models to model periodic patterns can improve language modeling performance, the underlying mechanisms responsible for this improvement remain underexplored. To the best of our knowledge, it has hardly been studied the role of periodicity or the potential periodic behaviors of LLMs on language modeling. Therefore, in future work, we will conduct a more comprehensive investigation into the fundamental mechanisms of periodicity in language modeling.

