# OpenReview forum: "Reasoning is Periodicity? Improving Large Language Models Through Effective Periodicity Modeling"
_NeurIPS.cc/2025/Conference — NeurIPS 2025 poster_

### Official Review · Reviewer_CN5c · 2025-06-25

**Clarity:** 4
**Significance:** 4
**Originality:** 4
**Rating:** 6
**Confidence:** 3

**Summary:**

The paper introduces a periodic component into transformer attention layer. The change is inspired by recent Fourier based network development as well as backed by the argument of periodicity importance (motivation section 2). After describing the network changes, the authors test the architecture on LM benchmarks against the natural baselines with controlled variations (4.4.1) as well as language models with similar size and capacities. The model achieves clear superior performance compared to baselines on common sense reasoning, language modelling as well as controlled math tasks and logical reasoning.

The paper tries to find elements that can explain the improvements: the paper shows the advantage of learning curve as well as the scaling properties. Finally, the paper also shows the model expressiveness advantage measured by Lipschitz measure.

**Questions:**

1. Do authors have tested different p value for the architecture. when p = 0 it goes to a linear layer and one can also only have fourier layers so it seems interesting to see what this parameter balances in terms of downstream performance as well as characteristics.

2. Do authors test on other benchmarks? As authors have mentioned, the improving mechanism is unknown so it is possible that other domains also simply improve and degradations are very useful to get noticed.

3. It is surprised to see that the model only gets pretrained but the performance exceeds some models that are fine-tuned or distilled. Do authors have comments on this please?

**Ethical Concerns:**

["NO or VERY MINOR ethics concerns only"]

**Final Justification:**

I would continue to recommend the paper as "strongly accept".

While I agree with the reviewers and authors that there is no clear motivation why periodicity modelling brings performance. I think the phenomenon and the modelling itself is exciting enough for the community given how popular transformer architecture is. It shows a modelling area that brings further fruits in a currently thought "mature" technology and architecture which bears high impact.

**Limitations:**

yes

**Quality:**

4

**Strengths And Weaknesses:**

Strength: Despite that the change is not straightforward, the paper is easy to follow where authors motivate the approach and explain clearly the changes that are brought to the existing architecture. The paper not only mentions the relevant work in the fourier network field but also connects with other attention improvement where the paper discusses the complementary roles of the current approach. The paper has extensive evaluations well organised. In terms of performance, the proposed model is evaluated against its natural baselines as well as compared with state-of-the-art models of similar size and show convincing improvements. Beyond that, the paper also makes a step to try to investigate where comes the improvement. Although the exact mechanism is still very difficult to find, the paper investigates into training curve, controlled task, scalability as well as representational power, laying a solid ground for future research to be built on top of the existing work. The work makes significant improvement on the seemingly well matured transformer architecture, so the contribution I judge to be significant to the field as well.

Weakness: There are some natural questions that I have when reading the paper. The paper can be even more extensive although it is definitely good on the thoroughness for the experiments.

---

> ### Author Rebuttal · Authors · 2025-07-31
>
> Thank you for your in-depth review and valuable feedback. We will respond to your comment point-by-point below.
>
> > Do authors have tested different p value for the architecture. when p = 0 it goes to a linear layer and one can also only have fourier layers so it seems interesting to see what this parameter balances in terms of downstream performance as well as characteristics.
>
> Yes, we present results for different p-values in Appendix C. But we fixed p = 0.25 and used the same setting in our experiments by default, which achieves stable and clear improvements. The experimental results in Figure 8 and Figure 9 of Appendix C show that, regardless of the variation in p values, 1) FANformer exhibits strong robustness in terms of training loss and downstream task accuracy, with relatively small performance fluctuations; 2) FANformer consistently outperforms the baselines. These findings indicate that FANformer maintains good robustness with respect to the hyperparameter p.
>
> As you point out, p=0 is an interesting baseline comparison, and the model degenerates to a purely linear layer at this value. The results for p=0, shown in Table 2 (i.e., the baseline ATL mentioned in this paper), perform significantly worse than FANformer.
>
> > Do authors test on other benchmarks? As authors have mentioned, the improving mechanism is unknown so it is possible that other domains also simply improve and degradations are very useful to get noticed.
>
> Thanks for your questions. We conduct additional experiments on code generation tasks (i.e., HumanEval and MBPP) and human-written math tasks (i.e., GSM8K) compared to our baseline, OLMo (which is the Outstanding Paper at ACL 2024),  and have added the experimental results to our revised manuscript. The results show that our FANformer achieves clear and consistent improvements compared with OLMo on all three benchmarks.
>
> Moreover, we conduct systematic testing on 35 language modeling tasks, and FANformer also achieves notable and stable improvements across these tasks compared to OLMo and the variants of FANformer, as shown in Table 6 and Table 7 of Appendix.
>
> | LLMs | Training Tokens | HumanEval | MBPP | GSM8K |
> |-|-|-|-|-|
> | OLMo-1B | 3T checkpoint  | 5.2     | 3.1       | 8.9        |
> | FANformer-1B  |  1T from scratch   | **6.3**     | **5.4**       | **15.7** |
> ||
>
>
> > It is surprised to see that the model only gets pretrained but the performance exceeds some models that are fine-tuned or distilled. Do authors have comments on this please?
>
> That's a great point. While techniques like fine-tuning and distillation are powerful for adapting a model to a specific task, they often come with the risk of the phenomenon called catastrophic forgetting or knowledge degradation, i.e, the model may lose some of the general capabilities acquired during pre-training. This underscores the critical importance of research on fundamental model architecture to improve learning efficiency.

---

> ### Author Response · Authors · 2025-08-09
>
> Dear Reviewer CN5c,
>
> Thank you very much for your detailed review, insightful comments, and strong support for our work. Your recognition and encouragement mean a great deal to us. Thank you once again for your valuable time and professional guidance!
>
> Best regards,
> Authors

---

### Official Review · Reviewer_GTjC · 2025-06-26

**Clarity:** 2
**Significance:** 3
**Originality:** 4
**Rating:** 4
**Confidence:** 3

**Summary:**

This paper introduces a new method that integrates the Fourier basis into the attention module, primarily motivated by the periodicity in human data (as they claim). They demonstrated that transformers with this periodicity component outperform alternative transformers that lack this architecture. They did ablation studies to show that the benefit primarily stems from the periodicity component. They conducted extensive experiments comparing their method to existing transformers on a range of datasets, including those that incorporate reuse. They showed that FANformer scales up, performs okay, and takes fewer parameters.

**Questions:**

From Section 4.4.3, it appears that FAN introduces more holes in unwanted regions around 99/98 in the linear regression task (figure 5 right most plot). Does FANFormer introduce new problems that transformers do not have?

There are a small number of holes in the modular addition task, but why does the transformer have holes shaped like a square?

Equation 1: what is p with an overline and p without an overline?

How is the tokenization done? Why does FANformer need a smaller number of training tokens?

The authors observe that effective periodicity modeling improves the learning efficiencies of LLMs. Does it induce loss in some other aspect, mainly when knowledge is not described in a periodic form?

How are the Lipschitz constants evaluated?

Line 27:
“Transformer-based models are also known for their immense demand for data and computational resources during training [Kaplan et al., 2020; Hoffmann et al., 2022; Chowdhery et al., 2023]. In comparison, humans can accomplish similar learning tasks with far fewer resources. This discrepancy suggests that existing LLM architectures still suffer from low learning efficiency.”
This is a valid observation, but it remains unclear whether the proposed FAN model addresses this gap in learning efficiency. The connection between periodicity and improved learning efficiency is not substantiated in the paper.

**Ethical Concerns:**

["NO or VERY MINOR ethics concerns only"]

**Final Justification:**

This paper proposes a novel architecture and shows that in some domains, the new architecture performs better than the transformer-based architecture in a number of evaluation experiments. Although the motivations for periodicity is a bit of a stretch, the new architecture would be a contribution to the community and I would recommend an acceptance.

**Limitations:**

The authors included limitations, but a discussion with other large models that take periodicity into account is currently lacking.

**Paper Formatting Concerns:**

I did not notice a formatting concern.

**Quality:**

3

**Strengths And Weaknesses:**

__Strengths__

The paper's strength lies in its extensive experiments and evaluations on a wide range of datasets, as well as investigations into the influence of hyperparameters on the loss. The organization and structure of the paper is also very clear.

__Weaknesses__

What is currently lacking is a comparison with alternative models that also employ periodicity in their architecture, which have also claimed that these models perform better than transformers, as it is not the first paper to find ways to cook periodicity into its architecture. For example, how does Fanformer compare to state space models (SSMs), such as mamba, with recurrence that naturally support periodic dynamics?

I found the motivation for introducing periodicity rather unconvincing.
The paper claims that periodicity facilitates structured knowledge acquisition. However, this is conceptually vague and poorly grounded. Knowledge, particularly in domains like language, does not naturally exhibit periodic structure. While one might metaphorically describe some mental representations as "wave-like," structured knowledge is not inherently represented as waveforms.

Line 31:
The authors cite Lake et al. and Buzsáki (2006) as part of the motivation for periodicity. However, these references tangentially relate to the claim. Lake et al. emphasize human learning efficiency through intuitive theories and compositional representations, periodicity is never mentioned. Buzsáki’s work hypothesizes that rhythmic activity underlies some brain functions, but this remains speculative. Furthermore, most supporting evidence pertains to animal studies and does not directly suggest that periodicity is a principle governing human learning processes, especially not in high-level domains like language.

Moreover, pattern recognition—which is a plausible explanation for human generalization—is not equivalent to with periodicity.
Lastly, it remains completely unclear how periodicity should manifest in language modeling. Mathematical functions like x mod 5 may be periodic, but this kind of structure is far removed from the kind of compositional and hierarchical patterns present in natural language.

---

> ### Author Rebuttal · Authors · 2025-07-31
>
> Thank you for your thorough review and detailed feedback. We will respond to your comment point-by-point below.
>
> > How does Fanformer compare to state space models (SSMs), such as mamba, with recurrence that naturally support periodic dynamics?
>
> First, our approach is fundamentally different from SSMs. SSMs model periodicity along the sequence dimension, while our FANformer models it along the feature dimension. Second, our motivation is distinct from that of Mamba. Mamba is primarily developed to overcome the quadratic computational complexity of Transformers and improve inference efficiency, while our approach is designed to improve learning efficiency and performance of Transformers through effective periodicity modeling.
>
>
> > From Section 4.4.3, it appears that FAN introduces more holes in unwanted regions around 99/98 in the linear regression task (figure 5 right most plot). Does FANFormer introduce new problems that transformers do not have?
>
> Sorry for the confusion, we find that although the test metric on the training set was close to 1, there is still a small amount of data that was not fitted when displayed. We continue to extend the training and find that this problem can be effectively solved. The final conclusion does not change. Transformer still shows overfitting on the training set, while FANformer facilitates the rule-based reasoning paradigm, mitigating the occurrence of "holes" inherent in the case-based learning of Transformer. Moreover, we have not yet observed specific scenarios where FANformer performs poorly. We leave this for our future work.
>
> > There are a small number of holes in the modular addition task, but why does the transformer have holes shaped like a square?
>
> The square shape of the hole in Transformer is due to the experimental setting. Following work [1], we randomly dug out the middle square part in the training set and used it as the test set, and the rest was used as the training set.
>
> Reference:
> [1] Case-based or rule-based: How do transformers do the math?  ICML 2024
>
> > Equation 1: what is p with an overline and p without an overline?
>
> The input X is multiplied by a weight matrix W, which is conceptually split into two parts: W_p and W_p̄. The part of the output corresponding to W_p (i.e., W_p X) is fed into both cos and sin functions to create periodic features, while W_p̄ undergoes a linear transformation. The overline notation signifies the remaining part of the feature transformation.
>
>
> > How is the tokenization done? Why does FANformer need a smaller number of training tokens?
>
> Tokenization is performed using a standard LLM tokenizer. We have not made any special modifications or optimizations to this process. In the standard pre-training process for LLMs, a massive text corpus is segmented into sequences of tokens, which are then fed into the model for training. The total number of tokens processed by the model is the precise measure of the training data used. In our comparative experiments (Figure 1, right), we observed that FANformer uses significantly fewer total training tokens to reach the same performance level as the traditional Transformer baseline. Therefore, our statement that FANformer "requires fewer training tokens" is a direct description of this experimental result. It demonstrates that FANformer has higher learning and data efficiency, achieving comparable or even better results with less training data. This proves the advantage of our new architecture in improving learning efficiency.
>
>
> > The authors observe that effective periodicity modeling improves the learning efficiencies of LLMs. Does it induce loss in some other aspect, mainly when knowledge is not described in a periodic form?
>
> This is a good question! Our current results suggest that introducing periodic modeling improves performance compared to existing models, but whether this will lead to performance loss is unknown. However, we have not yet observed specific scenarios where FANformer performs poorly. From our perspective, we retain the existing modeling approach while also adding frequency-domain modeling. This allows for some signals to be converted to the frequency domain. We believe this approach will not result in excessive information loss, but will compensate to some extent for the lack of periodic signals.
>
> > How are the Lipschitz constants evaluated?
>
> The Lipschitz constant L is defined as $\forall x, y \in \mathbb{R}^n, \quad \lVert f_{\text{model}}(x) - f_{\text{model}}(y) \rVert \leq L \lVert x - y \rVert$. We approximate L using a small perturbation $\epsilon=1e-7$, computing $\lVert f_{\text{model}}(x)-f_{\text{model}}(x+\epsilon) \rVert$. We perform 10 calculations and take the average to approximate L. The reproducible code of experiments is provided in the anonymous code project.
>
> > Line 27: “Transformer-based models are also known for their immense demand for data and computational resources during training [Kaplan et al., 2020; Hoffmann et al., 2022; Chowdhery et al., 2023]. In comparison, humans can accomplish similar learning tasks with far fewer resources. This discrepancy suggests that existing LLM architectures still suffer from low learning efficiency.” This is a valid observation, but it remains unclear whether the proposed FAN model addresses this gap in learning efficiency. The connection between periodicity and improved learning efficiency is not substantiated in the paper.
>
> Thanks for your comment. We demonstrate the connection between periodicity and improved learning efficiency through the following results. FANformer effectively improves the periodicity modeling performance, and FANformer also improves learning efficiency as evidenced by: 1) FANformer-1B outperforms an open-source LLM of similar size with fewer training tokens, and even surpasses an LLM with three times the number of parameters using the same training tokens. 2) FANformer consistently outperforms the Transformer, achieving comparable performance with only 69.2% of the model parameters or 79.7% of the training tokens. 3) By observing the training process, we find that FANformer's learning efficiency significantly improves compared to the Transformer as the model continuously learns from the data.
>
> These results, i.e, outperforming other models with fewer resources, demonstrating superior scaling-law performance, and achieving faster convergence, collectively confirm that FANformer's effective periodicity modeling is the key mechanism driving these measurable improvements in learning efficiency.

---

> > ### Comment · Reviewer_GTjC · 2025-08-03
> >
> > Thank you to the authors for their detailed and thoughtful rebuttal. Provided that the related literature is incorporated—particularly clarifying the distinction between FANformer and state space models (feature-wise vs. sequence-wise periodicity)—and the visualization in Section 4.4.3 is corrected, I would recommend this paper for acceptance.

---

> ### Author Response · Authors · 2025-08-04
>
> Reviewer GTjC,
>
> Thank you very much for your constructive feedback and your recommendation for acceptance of our work. We fully agree with your points and will further make two key revisions in the final version: 1) clearly state the core distinction between FANformer and SSMs, i.e., FANformer models periodicity along the feature dimension (feature-wise), whereas SSMs focus on the sequence dimension (sequence-wise), and 2) correct the visualization in Section 4.4.3.
>
> We sincerely appreciate the time and effort you have devoted to reviewing our work.
>
> Best regards,
> Authors

---

### Official Review · Reviewer_BuMP · 2025-07-01

**Clarity:** 3
**Significance:** 3
**Originality:** 3
**Rating:** 4
**Confidence:** 3

**Summary:**

This paper introduces FANformer, a novel large language model (LLM) architecture that improves learning efficiency and performance by enhancing the Transformer attention mechanism with periodicity modeling via Fourier Analysis Network (FAN). The authors argue that periodic patterns, ubiquitous in human reasoning, are poorly modeled by standard Transformers. FANformer modifies the attention mechanism to operate partially in the frequency domain using sine and cosine projections, resulting in better representation of periodic features. Extensive experiments show FANformer achieves comparable or superior performance to baseline Transformers with significantly fewer parameters and training tokens. It also demonstrates better rule-based reasoning, logical inference capabilities across several downstream tasks.

**Questions:**

1. Explanation of Validity
Authors should provide more detailed explanations as to why periodic modeling can enhance the capability of reasoning and why it can improve the efficiency of learning, rather than merely illustrating through experiments. Intuitively, modeling in the frequency domain would decrease the model efficiency.

2. Limited Scope of Evaluation Benchmarks
This paper primarily evaluates standard common-sense question-answering tasks and synthetic mathematical reasoning datasets, lacking systematic testing in more real-world task scenarios.

3. Real-world Applicability:
 Have you benchmarked the inference speed and GPU memory usage of FANformer relative to Transformer in deployment settings? Would the frequency-domain calculations add significant latency?

4. Generalization Beyond Commonsense Tasks:
Have you evaluated FANformer in other structured or periodic domains (e.g., code generation) where periodicity may be even more prevalent?

5. How to balance the model's capability in periodic modeling and its ability in non-periodic task modeling?

**Ethical Concerns:**

["NO or VERY MINOR ethics concerns only"]

**Limitations:**

YES

**Quality:**

2

**Strengths And Weaknesses:**

Innovation of Approach:
The use of a modified Fourier-based projection within the QKV structure of attention (the ATF module) is an original solution. By embedding periodic patterns directly into attention computation, the model diverges meaningfully from standard Transformer variants. While previous work explored Fourier Analysis Network, this paper proposes a principled and scalable adaptation.

Significance:
This work has high potential impact. Periodicity is underexplored in LLMs, and this approach opens a novel and elegant path for improving Transformer-based architectures, especially in domains where pattern repetition and rule-based reasoning are essential.

Clarity and Fluency:
The writing is generally clear, and the motivations are well articulated. However, a few sections (e.g., the mathematical formalism in Section 2) may be too dense for readers unfamiliar with group theory or frequency-domain analysis. Pseudocode and diagrams are helpful but could be further clarified.

Weaknesses:
1. The theoretical advantages of periodic modeling were not explained in further detail.
2. While promising, the reliance on sinusoidal expansions could introduce constraints in non-periodic linguistic domains. Furthermore, there is insufficient analysis of the model’s behavior in handling irregular or non-periodic sequences — a potential limitation for broad NLP applicability. While generalization is claimed, more rigorous checks (e.g., across languages and tasks) would have bolstered the argument.
3.  While periodic tasks are ideal tests for FANformer, real-world linguistic periodicity is more nuanced; extrapolation from synthetic setups (e.g., modular arithmetic) to natural language warrants caution.
4. Lack of Cross-Domain Validation– All tasks use English and a narrow band of benchmarks. Cross-linguistic or multimodal validation is absent.

---

> ### Author Rebuttal · Authors · 2025-07-31
>
> Thank you for your thorough review and detailed feedback. We will respond to your comment point-by-point below.
>
> > Generalization Beyond Commonsense Tasks: Have you evaluated FANformer in other structured or periodic domains (e.g., code generation) where periodicity may be even more prevalent? & Limited Scope of Evaluation Benchmarks This paper primarily evaluates standard common-sense question-answering tasks and synthetic mathematical reasoning datasets, lacking systematic testing in more real-world task scenarios.
>
> Thanks for your suggestions. We conduct experiments on code generation tasks (i.e., HumanEval and MBPP) and human-written math tasks (i.e., GSM8K) compared to our baseline, OLMo (which is the Outstanding Paper at ACL 2024),  and have added the experimental results to our revised manuscript. The results show that our FANformer achieves clear and consistent improvements compared with OLMo on all three benchmarks.
>
> Moreover, we conduct systematic testing on 35 language modeling tasks, and FANformer also achieves notable and stable improvements across these tasks compared to OLMo and the variants of FANformer, as shown in Table 6 and Table 7 of Appendix.
>
> | LLMs | Training Tokens | HumanEval | MBPP | GSM8K |
> |-|-|-|-|-|
> | OLMo-1B | 3T checkpoint  | 5.2     | 3.1       | 8.9        |
> | FANformer-1B  |  1T from scratch   | **6.3**     | **5.4**       | **15.7** |
> ||
>
> > Explanation of Validity Authors should provide more detailed explanations as to why periodic modeling can enhance the capability of reasoning and why it can improve the efficiency of learning, rather than merely illustrating through experiments. Intuitively, modeling in the frequency domain would decrease the model efficiency.
>
> Many real-world features are inherently periodic. For example, general reasoning can be understood as a form of periodicity under group actions as analyzed in Section 2. However, existing LLMs lack dedicated approaches to process these periodic characteristics. Our Fanformer is designed to effectively model such features. For periodic latent features, effectively modeling them improves LLM's learning efficiency, as demonstrated in our experiments.
>
> > Real-world Applicability: Have you benchmarked the inference speed and GPU memory usage of FANformer relative to Transformer in deployment settings? Would the frequency-domain calculations add significant latency?
>
> Thanks for your questions. We conduct experiments on the inference speed and GPU memory usage of FANformer relative to Transformer in deployment settings, and have added the experimental results to our revised manuscript. The results show that it adds little latency.
>
> The configuration of the benchmark test: we run for 20 iterations on a single GPU of A100 80G with a fixed sequence length of 4096 tokens and float16 precision.
>
> | Metric              | OLMo-1B     | FANformer-1B | Difference               |
> |---------------------|-------------|--------------|---------------------------|
> | Forward Pass Time   | 141.49 ms   | 142.88 ms    | +1.39 ms (+0.98%)        |
> | Allocated Memory    | 4642.69 MB  | 4738.86 MB   | +96.17 MB (+2.1%)         |
> | Peak Memory         | 6610.70 MB  | 6706.88 MB   | +96.18 MB (+1.5%)        |
> ||
>
>
> > How to balance the model's capability in periodic modeling and its ability in non-periodic task modeling?
>
> The periodicity ratio hyperparameter can be used to balance the model's capability in periodic modeling and non-periodic task modeling. In this paper, we fixed the periodicity ratio to 1/4 and used the same setting in all experiments, and achieved stable and clear improvements. The experimental results in Figure 8 and Figure 9 of Appendix show that, regardless of the variation in p values, 1) FANformer exhibits strong robustness in terms of training loss and downstream task accuracy, with relatively small performance fluctuations; 2) FANformer consistently outperforms the baselines. These findings indicate that FANformer maintains good robustness with respect to the hyperparameter p.

---

> > ### Comment · Reviewer_BuMP · 2025-08-05
> >
> > Thank you to the authors for their response; I recommend accepting this paper.

---

> > > ### Author Response · Authors · 2025-08-05
> > >
> > > Dear Reviewer BuMP,
> > >
> > > Thank you very much for your positive feedback and for recommending the acceptance of our paper. We are truly grateful for your support. Your insightful comments during the review process were invaluable and have significantly helped us improve the quality of our manuscript.
> > >
> > > Thank you once again for your time and thoughtful consideration.
> > >
> > > Best regards,
> > > Authors

---

### Official Review · Reviewer_txRr · 2025-07-02

**Clarity:** 3
**Significance:** 3
**Originality:** 3
**Rating:** 5
**Confidence:** 3

**Summary:**

The paper's primary contribution is introducing periodicity as a mechanism to enhance language modeling and reasoning capabilities in large language models. The authors provide both theoretical foundations and practical implementations for this approach.

**Questions:**

--How to detech if a setting suits to use periodicity modeling? Is there a preliminary step to determine whether to introducting a FANformer vs a regular transformer?

----While the study demonstrates that enhancing a language model's ability to model periodic patterns improves performance, the underlying mechanisms responsible for this improvement remain mysterious. Why there is naturally a periodic structure in reasoning and generalization?

--Limited testing scope - the approach has only been tested on a few datasets.

--The scope of the claims may be limited by the experimental setup. It remains unclear how well this new method generalizes to other settings.

--Results may depend on implicit assumptions that need to be articulated.

**Ethical Concerns:**

["NO or VERY MINOR ethics concerns only"]

**Final Justification:**

I wish to maintain my ratings.

**Limitations:**

--Computational Resource Constraints: Due to limits in computational resources, the researchers were only able to pretrain FANformer-1B model. The result would be more convincing if it is observed on larger models.

--FANformer architecture is designed to be orthogonal to other existing approaches for revising the attention mechanism, meaning it can seamlessly incorporate them. However, in this work, only Flash Attention was incorporated for necessary acceleration. The exploration and integration of other attention variants, are left for future research.

**Quality:**

3

**Strengths And Weaknesses:**

Contribution and strengths:

--Incorporating Fourier transforms into Transformer architectures—and potentially future LLMs—is a challenging problem. On the other hand, it is well known that such integration could be a breakthrough for handling data with inherent periodicity. This paper explores new model architectures to address this challenge and demonstrates strong empirical performance."

--The paper includes a wide range of downstream tasks, which makes the proposed architecture particularly appealing.

--The authors provide detailed descriptions of their model architectures, hyperparameters, and training procedures, ensuring reproducibility of their experimental results.

--The paper also offers theoretical explanations for why periodicity facilitates language modeling and reasoning, supported by formal proofs of their methodological claims, and computational cost analysis.



Weaknesses:
--Computational Resource Constraints: Due to limits in computational resources, the researchers were only able to pretrain FANformer-1B model. The result would be more convincing if it is observed on larger models.

--FANformer architecture is designed to be orthogonal to other existing approaches for revising the attention mechanism, meaning it can seamlessly incorporate them. However, in this work, only Flash Attention was incorporated for necessary acceleration. The exploration and integration of other attention variants, are left for future research.

---

> ### Author Rebuttal · Authors · 2025-07-31
>
> Thank you for your in-depth review and valuable feedback. We will respond to your comment point-by-point below.
>
> > How to detech if a setting suits to use periodicity modeling? Is there a preliminary step to determine whether to introducting a FANformer vs a regular transformer?
>
> There is currently no precise method to determine if a setting is suitable for using periodicity modeling. We recommend considering FANformer as a more powerful default option to use.  Because various forms of real-world data contain either explicit or implicit periodic patterns, and we believe that these tasks will benefit from FANformer. When a task is completely aperiodic, FANformer essentially degenerates to a standard Transformer. To date, we have not yet identified specific scenarios where it performs poorly. However, the boundaries of FANformer's generalizability remain unknown, and we leave this exploration for future work.
>
>
> > While the study demonstrates that enhancing a language model's ability to model periodic patterns improves performance, the underlying mechanisms responsible for this improvement remain mysterious. Why there is naturally a periodic structure in reasoning and generalization?
>
> Thanks for your question. As described in Limitation Section, "although we have observed that enhancing the ability of language models to model periodic patterns can improve language modeling performance, the underlying mechanisms responsible for this improvement remain underexplored. To the best of our knowledge, it has hardly been studied the role of periodicity or the potential periodic behaviors of LLMs on language modeling. Therefore, in future work, we will conduct a more comprehensive investigation into the fundamental mechanisms of periodicity in language modeling." We will try to explain this question from two perspectives:
>
> First, based on the strict definition of periodicity, we will give some intuitive observations and explanations.
>
> 1. **The essence of periodicity lies in the repetitive manifestation of certain invariance under transformations**, which can be strictly defined through invariance under group actions in Abstract Algebra. *(Let $X$ be a set and $G$ be a group acting on $X$. An element $x \in X$ is said to be periodic with respect to the action of $G$ if there exists a non-identity element $p \in G$ such that $p \cdot x = x$, where $\cdot$ denotes the action of the group $G$ on the set $X$. The element $p$ is called a period of $x$ under the action of $G$. Periodicity manifests as the invariance of $x$ under all elements of the cyclic subgroup generated by $p$, denoted by $\langle p \rangle$. If the group operation in $G$ is written multiplicatively, then $\langle p \rangle =$ {$p^n \mid n \in \mathbb{Z}$}. For every element $g \in \langle p \rangle$, we have $g \cdot x = x$.)* For example, $f(a) = f(a+T)$ can be seen as a specific instance of the abstract definition $p \cdot x = x$, where $x=f$, $p=T$, and the group action is translation. When the input $a$ and the group $G$ are extended to higher dimensions or non-temporal domains, the manifestation of the period $T$ also changes accordingly. In simple terms, **for inputs belonging to the same domain (or having the same properties), using the same rule $f$ to calculate, its resulting output implies invariance. It's typical for most reasoning questions to take this form.**
> 2. Intuitive Explanation: Assuming that certain knowledge has already been learned within the semantic feature space, the emergence of new concepts benefits from the inherent periodicity by preferentially establishing connections with existing knowledge, rather than requiring de novo learning.
> 3. In Section 4.4.3, we experimentally explore that FANformer facilitates the rule-based learning paradigm of math reasoning in natural language, effectively mitigating the occurrence of "holes" inherent in the case-based reasoning of Transformer (Hu et al., 2024, ICML).  In Section 4.4.4, under the stress test of logical reasoning [Wang et al., 2024], FANformer-1B demonstrates superior performance compared to OLMo-1B and Qwen2.5-1.5B.
>
> In summary, the core of periodicity lies in the recurrence of a system's certain rule under specific transformations, which is common and important in reasoning and language modeling.
>
> Second, for the generalization, we believe that some generalization properties can be expressed as periodicity. This is because many real-world features are inherently periodic, which makes it reasonable to generalize based on this characteristic. Furthermore, periodic extension is a valid and parsimonious assumption that complies with Occam's razor. But we must acknowledge that the extent of its generalization capability is likely highly dependent on the specific real-world task.
>
> > Limited testing scope - the approach has only been tested on a few datasets. & The scope of the claims may be limited by the experimental setup. It remains unclear how well this new method generalizes to other settings. & Results may depend on implicit assumptions that need to be articulated.
>
> Thank you for your valuable questions about test scope, generalization ability, and implicit assumptions. To ensure the rigor and comprehensiveness, our evaluation process follows the widely recognized previous work, especially the evaluation framework established by the work OLMo (Outstanding Paper at ACL 2024), which ensures that our experimental settings and benchmark selection are fair and representative. Specifically, we conduct experiments on 8 core tasks of commonsense in Table 1, 35 tasks of language modeling in Table 2, and 4 tasks of instruction following in Table 4. In addition to adhering to OLMo's evaluation framework, we further validated 1) scalability of FANformer in accordance with scaling laws in Figure 3, 2) generalization capability of FANformer on case-based and rule-based reasoning tasks in Figure 5 and Figure 10, 3) logical reasoning ability of FANformer-1B under the stress test in Table 3 and Figure 13. To further validate the model's generalization and practical effectiveness, we conduct additional experiments on code generation tasks (i.e., HumanEval and MBPP) and human-written math tasks (i.e., GSM8K) compared to our baseline, OLMo,  and have added the experimental results to our revised manuscript. The results show that our FANformer achieves clear and consistent improvements compared with OLMo on all three benchmarks.
>
> | LLMs | Training Tokens | HumanEval | MBPP | GSM8K |
> |-|-|-|-|-|
> | OLMo-1B | 3T checkpoint  | 5.2     | 3.1       | 8.9        |
> | FANformer-1B  |  1T from scratch   | **6.3**     | **5.4**       | **15.7** |
> ||

---

> > ### Comment · Reviewer_txRr · 2025-08-06
> >
> > I thank the authors for their efforts in providing explanations and observations in response to my questions, which help address some of my concerns and clarify several points.

---

> ### Author Response · Authors · 2025-08-06
>
> Dear Reviewer txRr,
>
> Thank you for taking the time to review our paper and provide helpful feedback. We are very pleased to hear that our response helped clarify several points and address some of your concerns. Your feedback has been valuable for improving our manuscript, and we truly appreciate the opportunity for this discussion. Thank you again!
>
> Best regards,
> Authors

---

### Note · Authors · 2025-08-16

Dear reviewers and chairs,

We thank all reviewers for their positive and insightful feedback, as well as their willingness to accept our paper. We are delighted that the reviewers recognized the value of our contributions. We would like to summarize the key strengths they highlighted:

 * Reviewer BuMP points out that "This work has high potential impact. Periodicity is underexplored in LLMs, and this approach opens a novel and elegant path for improving Transformer-based architectures, especially in domains where pattern repetition and rule-based reasoning are essential."

 * Reviewer txRr noted that "it is a challenging problem", "such integration could be a breakthrough for handling data with inherent periodicity", and "The paper also offers theoretical explanations for why periodicity facilitates language modeling and reasoning, supported by formal proofs of their methodological claims, and computational cost analysis".

* Reviewer CN5c summarized that "The work makes significant improvement on the seemingly well-matured transformer architecture, so the contribution I judge to be significant to the field as well".

* All reviewers lauded the "strong empirical performance" and the "extensive experiments and evaluations", which were described as "well organised" and "convincing". The paper was generally praised for being well-structured, clear, and easy to follow.

We are pleased that our rebuttal addressed the concerns raised by the reviewers. As suggested, we have incorporated the clarifications and revisions in our response into the final version of the paper. Thanks very much for the time and effort invested by all the reviewers and chairs in evaluating our work.

Best regards,
Authors

---

### Decision · Program_Chairs · 2025-09-17

**Decision:**

Accept (poster)

**Comment:**

This paper introduces a Transformer variant, FANformer, that integrates Fourier-based periodicity modeling into attention. Empirical studies  show that FANformer achieves comparable performance (both on pretraining metrics as well as on some downstream tasks) with fewer parameters or fewer training tokens than baseline Transformers. As motivation it is claimed that standard transformers poorly capture periodic patterns in reasoning and language modeling, however, the sinusoidal transformation is applied on feature dimension and not along token axis which makes the argument unclear. Nevertheless, all the reviewers leaned positive, citing novelty, clarity, and strong empirical evidence. Main concerns focused on limited scale (experiments at ~1B parameters), generalization to broader, and motivation/explanation of why adding the sinusoidal terms is a good idea. We thank the reviewers and authors for engaging during the rebuttal period to improve the paper. The rebuttal provided additional evaluations (HumanEval, MBPP, GSM8K) and clarifications that strengthened the paper. The two reviewers (BuMP and GTjC) with borderline score also explicitly recommend accepting this paper in final response. Overall, despite unclear motivation, the work demonstrates meaningful improvements over mature Transformer baselines and thus should be accepted.